# Necdin shapes serotonergic development and SERT activity modulating breathing in a mouse model for Prader-Willi syndrome

Valéry Matarazzo[1]*, Laura Caccialupi[1†], Fabienne Schaller[1†], Yuri Shvarev[2†], Nazim Kourdougli[1], Alessandra Bertoni[1], Clément Menuet[1], Nicolas Voituron[3], Evan Deneris[4], Patricia Gaspar[5], Laurent Bezin[6], Pascale Durbec[7], Gérard Hilaire[1], Françoise Muscatelli[1]*

[1]Aix Marseille Univ, INSERM, INMED, Marseille, France; [2]Department of Women's and Children's Health, Karolinska Institute, Solna, Sweden; [3]Université Paris 13, UFR STAPS, Paris, France; [4]Department of Neurosciences, Case Western Reserve University, Cleveland, United States; [5]UPMC Univ Paris 6, Institut du Fer à Moulin, INSERM, Paris, France; [6]Lyon Neuroscience Research Center, Université de Lyon, INSERM, CNRS, Lyon, France; [7]Aix Marseille Univ, CNRS, IBDM, Marseille, France

**Abstract** Prader-Willi syndrome (PWS) is a genetic neurodevelopmental disorder that presents with hypotonia and respiratory distress in neonates. The *Necdin*-deficient mouse is the only model that reproduces the respiratory phenotype of PWS (central apnea and blunted response to respiratory challenges). Here, we report that *Necdin* deletion disturbs the migration of serotonin (5-HT) neuronal precursors, leading to altered global serotonergic neuroarchitecture and increased spontaneous firing of 5-HT neurons. We show an increased expression and activity of 5-HT Transporter (SERT/Slc6a4) in 5-HT neurons leading to an increase of 5-HT uptake. In *Necdin*-KO pups, the genetic deletion of *Slc6a4* or treatment with Fluoxetine, a 5-HT reuptake inhibitor, restored normal breathing. Unexpectedly, Fluoxetine administration was associated with respiratory side effects in wild-type animals. Overall, our results demonstrate that an increase of SERT activity is sufficient to cause the apneas in *Necdin*-KO pups, and that fluoxetine may offer therapeutic benefits to PWS patients with respiratory complications.
DOI: https://doi.org/10.7554/eLife.32640.001

*For correspondence:
valery.matarazzo@inserm.fr (VM);
francoise.muscatelli@inserm.fr
(FM)

†These authors contributed
equally to this work

**Competing interests:** The
authors declare that no
competing interests exist.

**Reviewing editor:** Joseph G
Gleeson, Howard Hughes
Medical Institute, The Rockefeller
University, United States

## Introduction

Respiration is a complex function controlled in large part by raphe serotonergic (5-HT) neurons (*Teran et al., 2014*). Central 5-HT depletion induces severe apneas during the early postnatal period (*Barrett et al., 2016*; *Trowbridge et al., 2011*) and serotonopathy is implicated in the genesis of breathing disorders in human pathologies including neurodevelopmental diseases such as Sudden Infant Death Syndrome (*Duncan et al., 2010*; *Hilaire et al., 2010*; *Kinney et al., 2011*; *Paterson et al., 2009*), Rett syndrome (*Abdala et al., 2010*; *Toward et al., 2013*) and Prader-Willi Syndrome (PWS) (*Zanella et al., 2008*). However, the cellular and molecular events that underlie serotonopathy, and the causal link between serotonopathy and respiratory dysfunction in these pathologies are poorly understood.

PWS (prevalence 1/20000) is characterized by a combination of endocrine, metabolic, cognitive and behavioural/psychiatric symptoms (OMIM #176270). Its associated respiratory disturbances (*Miller and Wagner, 2013*; *Nixon and Brouillette, 2002*; *Tan and Urquhart, 2017*) are highly disruptive to the daily life of patients and represent the most common cause of death (73% of infants and 26% of adults) (*Butler et al., 2017*). They include both obstructive (*Festen et al., 2006*;

**eLife digest** Prader-Willi syndrome results from the disruption of a cluster of neighboring genes, including one called Necdin. Symptoms begin in early infancy and worsen with age. Affected children tend to develop an insatiable appetite, which often leads to obesity. They also experience serious problems with their breathing. Chest infections, high altitude and intense physical activity can be dangerous for children with Prader-Willi syndrome. This is because a slight shortage of oxygen may trigger breathing difficulties that could prove fatal.

The brain cells that produce a chemical messenger called serotonin help to control breathing. Several lines of evidence suggest that loss of Necdin may trigger breathing difficulties in Prader-Willi syndrome via effects on the serotonin system. First, serotonin neurons produce the Necdin protein. Second, laboratory mice that lack the gene for Necdin have abnormally shaped serotonin neurons. Third, these mice show breathing difficulties like those of individuals with Prader-Willi syndrome. But while this implies a connection between serotonin, Necdin and breathing difficulties, it falls short of establishing a causal link.

Matarazzo et al. now reveal an increase in the quantity and activity of a protein called the serotonin transporter in mutant mice that lacked the gene for Necdin compared to normal mice. Serotonin transporter proteins mop up the serotonin that neurons release when they signal to one another. Neurons in the mutant mice take up more serotonin than their counterparts in normal mice; this means they have less serotonin available for signaling. This may make it harder for the mutant mice to regulate their breathing.

Drugs called selective serotonin-reuptake inhibitors (or SSRIs for short) can block the serotonin transporter. These drugs, which include Fluoxetine (also called Prozac), are antidepressants. Matarazzo et al. show that SSRIs temporarily restore normal breathing in young mice that lack the gene for Necdin. However, these drugs have harmful long-term effects on breathing in non-mutant mice. Further studies should test whether short-term use of SSRIs could offer immediate relief for breathing difficulties in infants and children with Prader-Willi syndrome.

DOI: https://doi.org/10.7554/eLife.32640.002

---

Pavone et al., 2015) and central sleep apneas {Festen et al., 2006 #1495; Sedky et al., 2014), and blunted responses to hypercapnia/hypoxia possibly due to a lack of chemoreceptor sensitivity (Arens et al., 1996; Gozal et al., 1994; Schlüter et al., 1997; Gillett and Perez, 2016). Central apneas are present at birth (Zanella et al., 2008) and are prevalent throughout infancy while obstructive sleep apneas are more frequent in adolescents (Cohen et al., 2014).

PWS is caused by the loss of paternal expression of several genes of the 15q11-q13 region, including NECDIN. Necdin protein is a member of the Mage family, with proposed functions in differentiation (Andrieu et al., 2003; Takazaki et al., 2002), migration (Kuwajima et al., 2010; Miller et al., 2009; Tennese et al., 2008), neurite growth (Liu et al., 2009; Tennese et al., 2008), axonal extension, arborization and fasciculation (Pagliardini et al., 2005), and cell survival (Aebischer et al., 2011; Andrieu et al., 2006; Kuwako et al., 2005; Tennese et al., 2008). Among several mouse models of PWS, only those with Necdin deletion, Necdin (Ndn)-KO mouse models (Ndn$^{tm1-Stw}$ [Gérard et al., 1999] and Ndn$^{tm1-Mus}$ [Muscatelli et al., 2000]), present breathing deficits. Newborns Ndn-KO showed severe arhythmia, apnea, and blunted responses to respiratory challenges that frequently result in early postnatal lethality (Ren et al., 2003; Zanella et al., 2008). This dyspnoeic phenotype is recapitulated in brainstem slices that contain the Inspiratory Rhythm Generator (IRG), which display an irregular inspiratory rhythm and apneas (Ren et al., 2003; Zanella et al., 2008). Interestingly, 5-HT application, as well as other neuromodulators that are commonly co-released by medullary 5-HT neurons, such as substance P and thyrotropin-releasing hormone (Hodges and Richerson, 2008; Holtman and Speck, 1994; Kachidian et al., 1991; Ptak et al., 2009), stabilized the in vitro inspiratory rhythm (Pagliardini et al., 2005; Zanella et al., 2008).

A role for serotonergic transmission in the genesis of respiratory dysfunction in the Necdin-KO model is supported by neuroanatomical studies: Pagliardini and colleagues report abnormal morphology and orientation of axonal fibers that contain large 5-HT/Substance P varicosities in the developing Ndn$^{tm1-Stw}$-KO medulla (Pagliardini et al., 2005; Pagliardini et al., 2008). Similarly, we

have also previously found that 5-HT fibers contained 'swollen 5-HT varicosities' in the $Ndn^{tm1-Mus}$-KO model, and that *Necdin* is expressed in virtually all 5-HT neurons (*Zanella et al., 2008*).

These findings suggest a potential role for abnormalities in 5-HT metabolism and release as a potential mediator of respiratory dysfunction in the *Necdin*-KO model of PWS, but fall short of proving causality. Here, we demonstrate a causal link between the perturbed development of the 5-HT system in $Ndn^{tm1-Mus}$-KO mice (referred to hereafter as *Ndn*-KO) and their observed respiratory phenotype (central apnea and hypercapnia). Our data implicate increased activity of serotonin transporter (SERT) as a key mediator of central apnea in this model, and that its inhibition restores normal breathing in *Ndn*-KO mice.

## Results and discussion

### Lack of Necdin affects the development and function of 5-HT neurons

Pet-EYFP mice expressing YFP under Pet1-promoter control, an early marker of developing 5-HT neurons (*Hawthorne et al., 2010*), were used to show that Necdin is expressed from E10.5 in early post-mitotic 5-HT precursors and later on in all 5-HT neurons until adulthood (*Figure 1A*, *Figure 1— figure supplement 1A–I*).

We then assessed whether Necdin deficiency could induce alterations of 5-HT neuronal development. In wild-type mice rostral hindbrain 5-HT neurons project to the mesencephalon at E12.5, and we observed a decrease in those ascending 5-HT projections in *Ndn*-KO embryos (*Figure 1—figure supplement 1J*), confirming previous work (*Pagliardini et al., 2008*). At E16.5, when the 5-HT raphe nuclei reach their mature configuration, we observed misplaced 5-HT neurons in *Ndn*-KO embryos (*Figure 1B*), with ~30% reduction in the total number of 5-HT neurons in the B1-B2 caudal raphe nuclei at birth (*Figure 1C*).

Our observations suggested a defect in 5-HT neuronal migration; which was tested using the *Pet-EYFP* model. In E10.5 WT embryos, Pet-EYFP neurons displayed typical bipolar morphology with oval-shaped somata aligned with two primitive processes attached to the ventricular and pial surfaces, required for somal translocation and involved in migration processes (*Hawthorne et al., 2010*) (*Figure 1D*). In contrast cells were not correctly aligned and process orientation was significantly disturbed in *Pet-EYFP/Ndn*-KO embryos (*Figure 1D–E*). Cell migration was also defective in organotypic slice cultures prepared from E12.5 embryos. Two-photon time-lapse imaging indicated that migratory behavior, based on somal translocation, was altered in *Ndn*-KO mice (*Figure 1F–H*, *Figure 1—video 1* and *2*) with tracked cells exhibiting increased tortuosity (*Figure 1G*) and decreased velocity (*Figure 1H*) of their growth trajectories. Interestingly, a comparable migration defect has been described in primary cultures of $Ndn^{tm1-Stw}$-KO cortical neurons (*Bush and Wevrick, 2010*) Here we revealed an alteration of cell migration of 5-HT precursors leading to misplaced 5-HT raphe nuclei in *Ndn*-KO mice.

The acquisition of specific firing properties is considered a critical marker of 5-HT neuronal and circuit maturation (*Rood et al., 2014*). Using visually guided patch-clamp recordings on brain slices (P15), we demonstrated a significant increase of spontaneous firing in *Pet-EYFP/Ndn*-KO cells (*Figure 1I–K*) suggesting a decreased availability of extracellular 5-HT (*Maejima et al., 2013*). Overall, our results show that Necdin is responsible for the normal migration of 5-HT precursor neurons during development and exerts effects on their electrophysiological properties in post-natal life.

### Lack of Necdin increases the expression and activity of serotonin transporter

We hypothesised that reduced availability of extracellular 5HT could have contributed to the excessive electrophysiological activity we observed in Pet-EYFP neurons in *Ndn*-KO animals and examined potential mechanisms through which extracellular 5-HT could be reduced. We compared the distributions of 5-HT- immunoreactive enLarged Punctiform Axonal stainings (5-HT LPAs, previously named 'swollen large varicosities' [*Pagliardini et al., 2005*; *Zanella et al., 2008*]) in *Ndn*-KO and WT mice. In all regions analyzed we found significantly more 5-HT LPAs in *Ndn*-KO mice (*Figure 2A–B*). These 5-HT LPAs could result from (1) an increase of 5-HT synthesis and/or (2) a decrease in 5-HT degradation and/or (3) an increase of 5-HT reuptake. HPLC analyses showed a similar level of L-Trp and 5-HT in *Ndn*-KO compared with WT mice, but a significant increase of 5HIAA product in

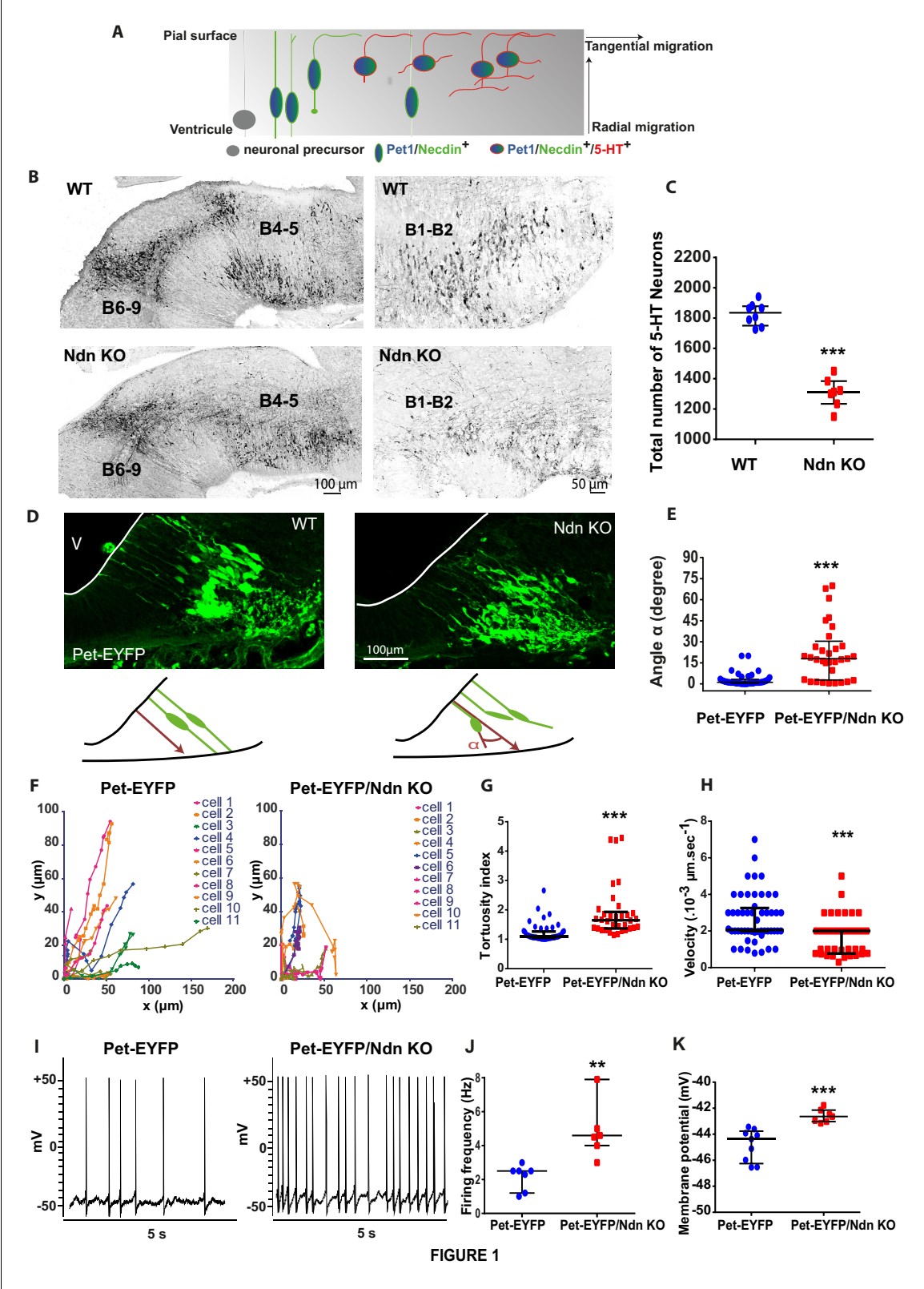

FIGURE 1

**Figure 1.** Necdin expression in 5-HT neurons and alterations of 5-HT neuronal development and activity in *Ndn*-KO mice. (A) Scheme adapted from (*Hawthorne et al., 2010*) representing expression profiles of Necdin (green), Pet1 (blue) and 5-HT (red) throughout embryonic development of 5-HT neurons as soon as the progenitors become post-mitotic and start their radial migration by successive waves between E11.5 and E13.5. (B–C) 5-HT immunolabelling of brainstem sagittal sections of WT and *Ndn*-KO at E16.5. (B) 5-HT nuclei: B4 to B9 (left panels) or B1-B2 (right panels) are abnormal

*Figure 1 continued on next page*

Figure 1 continued

in *Ndn*-KO compared to WT. (C) Quantification of 5-HT neurons in the B1-B2 raphe nuclei (WT: 1836 (1751, 1878), n = 8; *Ndn*-KO: 1312 (1234, 1384), n = 7; MW, p=0.0003). (D–E) (D) Brainstem coronal sections of *Pet-EYFP* neurons from E11.5 WT and *Ndn*-KO mice illustrating radial migration from the ventricular zone (V) to the pial surface. (E) Quantification of nonlinear migration by measuring the α angle (10 cells/mouse) between the ventricular process and a virtual axis crossing the two opposing points from which neurites extend from the soma: (Angle (°): WT: 1.1(0.5, 3.5), n = 4; *Ndn*-KO: 18 (2.8, 93.5), n = 3; MW, p<0.0001). (F–H) Confocal time-lapse analyses of cell migration of *Pet-EYFP* and *Pet-EYFP/Ndn*-KO neurons. (F) Plots representing the coordinates of individual cell bodies over time illustrate different cell migration patterns in WT (n = 4) and *Ndn*-KO (n = 3) *Pet-EYFP* neurons (11 cells/mouse). (G) Tortuosity index was increased by 52% in *Ndn*-KO compared to WT mice (WT: 1.08(1.01, 1.26); *Ndn*-KO: 1.65(1.36, 1.93); MW, p=0.0005). (H) Velocity was decreased by 37% in *Ndn*-KO compared to WT (Velocity ($\mu$m.s$^{-1}$): WT: 2.50 10$^{-3}$ (2.00, 2.93); *Ndn*-KO: 1.57 10$^{-3}$ (1.09, 2.00); MW, p<0.0001). (I–K) Current clamp recordings of *Pet-EYFP* neurons (2 cells/slice) in WT (n = 3) and *Ndn*-KO (n = 3) brain slices. (I) Spontaneous discharge pattern of *Pet-EYFP* neurons; (J–K) Firing rate (J) and resting membrane potential (K) in *Ndn*-KO cells and aged-matched WT controls. Frequency (Hz): WT: 2.50(1.20, 2.50); *Ndn*-KO (4.60(4.00, 7.90); MW, p=0.0025; Voltage (mV): WT: −44.37(-46.25, –43.76); *Ndn*-KO: −42.64(-43.03, –42.55); MW test, p=0.0002.

DOI: https://doi.org/10.7554/eLife.32640.003

The following video and figure supplement are available for figure 1:

**Figure supplement 1.** Necdin expression compared with Pet-1 and 5-HT expression throughout embryonic development and alteration of 5-HT projections in *Ndn*-KO embryos.

DOI: https://doi.org/10.7554/eLife.32640.004

**Figure 1—video 1.** Two-photon timelapse video showing somal translocation on organotypic slice cultures of Pet-EYFP neurons in WT embryos (E12.5).

DOI: https://doi.org/10.7554/eLife.32640.005

**Figure 1—video 2.** Two-photon timelapse video showing somal translocation on organotypic slice cultures of Pet-EYFP neurons in *Ndn*-KO embryos (E12.5).

DOI: https://doi.org/10.7554/eLife.32640.006

mutants (the ratio of 5HIAA/5-HT also being increased: *Figure 2—figure supplement 1A–D*). Noticeably, transcript levels of Tryptophan hydroxylase 2, the enzyme that converts L-Trp to 5-HT, were similar in *Ndn*-KO and WT mice (*Figure 2—figure supplement 1E*). These results suggest that the increase in 5-HT LPAs found in *Ndn*-KO brainstems probably result from an accumulation of intracellular 5-HT due to an increased 5-HT reuptake, since there is no increase of 5-HT synthesis but, on the contrary, an increase of 5-HT degradation.

We hypothesised that overexpression of serotonin transporter (SERT) represents a plausible mechanism through which 5-HT could be accumulated in *Ndn*-KO mice, based on the observation that inactivation of *Maged1*, another member of the *Mage* gene family, leads to overexpression of SERT (encoded by the *Slc6a4* gene) (*Mouri et al., 2012*). Indeed, we observed a 3.2 fold increase in SERT protein expression in the brainstems of *Ndn*-KO compared to WT pups (*Figure 2C–D*), while *Slc6a4* transcript levels were similar (*Figure 2—figure supplement 1F*). This suggests post-transcriptional or post-translational dysregulation of *Slc6a4*/SERT in *Ndn*-KO. Subsequently, in 5-HT neurons of raphe primary cultures, we assessed SERT activity by live single cell uptake assay, using ASP+ (4 (4-(dimethylamino)styryl)-N-methylpyridinium), a fluorescent substrate of SERT (*Lau et al., 2015*; *Oz et al., 2010*). Changes in the kinetics and saturation of ASP+ uptake were measured after 8 days in vitro culture in 5-HT neurons from neonatal (P0) WT, *Ndn*-KO, and *Slc6a4*-KO mice (*Figure 2E–H*, *Figure 2—figure supplement 2A–B*). As expected, cultures accumulated ASP+ over time in all conditions tested. However, kinetics experiments show that ASP+ accumulation was significantly faster (greater mean velocity v) in *Ndn*-KO compared to WT raphe neurons (*Figure 2E*). Saturation experiments using increasing concentrations of ASP+ confirmed that ASP+ uptake is a saturable process (*Figure 2F*) and showed a Vmax (*Figure 2G*) and KM (*Figure 2H*) significantly higher in *Ndn*-KO than in WT or *Slc6a4*-KO neurons. ASP+ uptake was ~2 fold increased in *Ndn*-KO while it was null in *Slc6a4*-KO cell cultures. We conclude that there is an increase of ASP+ uptake in *Ndn*-KO neurons, specifically dependent on SERT activity, suggesting a mechanism for 5-HT LPAs accumulation in vivo. To determine whether in vivo deletion of *Slc6a4* could suppress the 5-HT LPAs in *Ndn*-KO, we compared the number of 5-HT LPAs in *Ndn*-KO, *Slc6a4*-KO and *Ndn/Slc6a4*-double KO (*Ndn/Slc6a4*-DKO) neonates in various brain structures. The number of 5-HT LPAs was similar in brains of *Ndn/Slc6a4*-DKO and WT mice (*Figure 2A–B*), indicating that the absence of *Ndn* is functionally compensated for by the lack of *Slc6a4*.

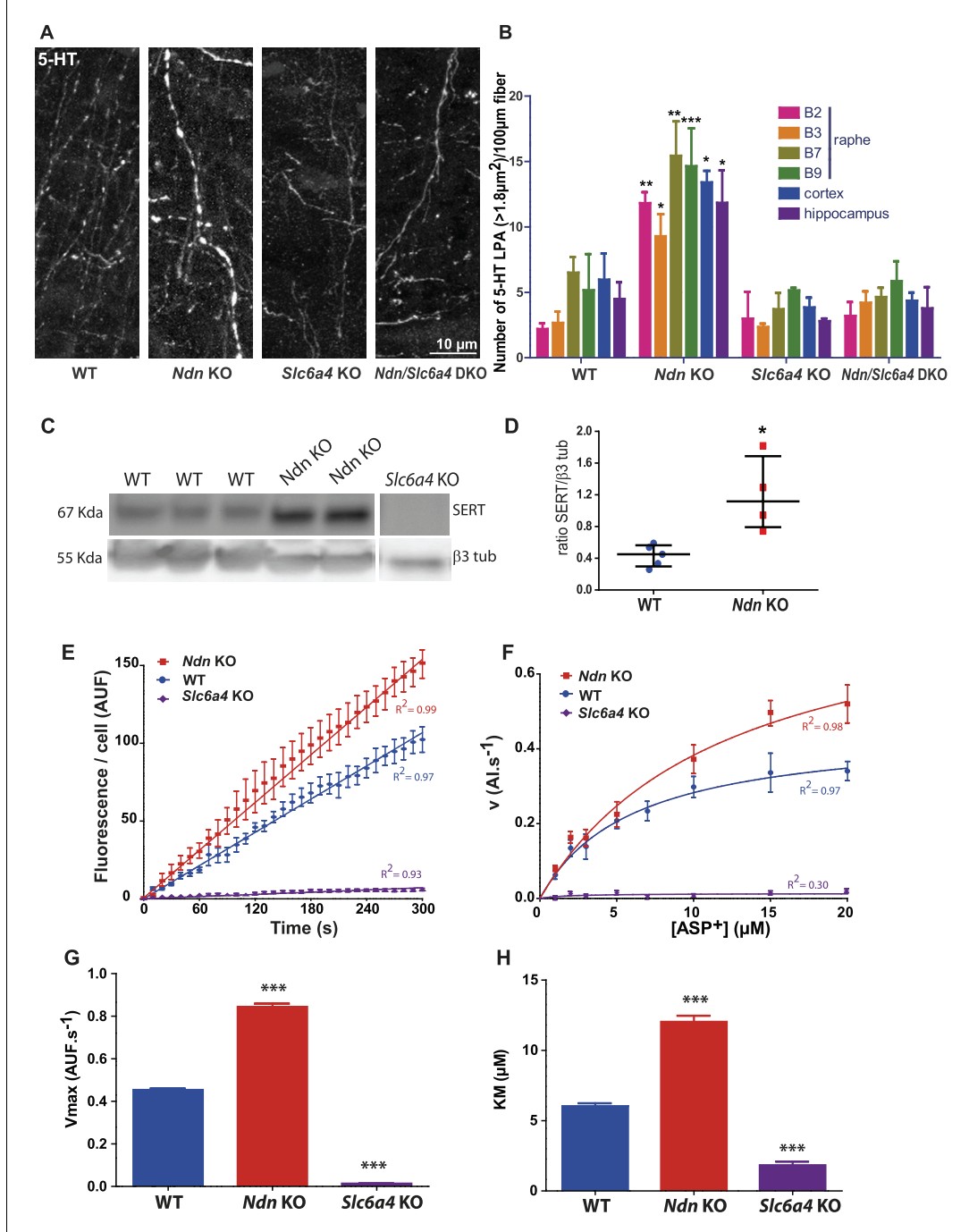

**Figure 2.** Large punctiform axonal 5-HT staining (5-HT LPAs) results from an increase in SERT expression and activity in *Ndn*-KO mice. (**A–B**) (**A**) Axonal 5-HT immunoreactivity illustrating 5-HT LPAs in the raphe of WT, *Ndn*-KO, *Slc6a4*-KO and *Ndn/Slc6a4*-DKO neonates (P1). (**B**) 5-HT LPAs were counted for all different genotypes (n = 3/genotype) in the raphe nuclei (B1–B2, B3, B7, B9), cortex and hippocampus.: Raphe B1-B2: WT: 2.2 ± 0.4; *Ndn*-KO: 11.8 ± 0.8, p=0.003; Raphe B3: WT: 2.6 ± 0.8; *Ndn*-KO: 9.3 ± 1.6, p=0.01; Raphe B7: WT: 6.5 ± 1.2; *Ndn*-KO: 15.5 ± 2.6, p=0.07; Raphe B9: WT: 5.1 ± 2.72; *Ndn*-KO: 14.6 ± 2.9, p=0.0001; Cortex: WT: 5.9 ± 2.0; *Ndn*-KO: 13.4 ± 0.8, p=0.01; Hippocampus: WT: 4.5 ± 1.2; *Ndn*-KO: 11.8 ± 2.5, p=0.01. p-values determined by two-way ANOVA followed by Bonferroni post-hoc test. DKO: double KO. Bar graphs represent mean ±SEM. (**C–D**) (**C**) Western blot analysis of SERT protein expression in brainstem collected from WT, *Ndn*-KO and *Slc6a4*-KO (negative control) neonates (P1). (**D**) Quantification of SERT expression normalized to β3 tubulin expression: WT: 0.45 (0.29, 0.56), n = 5; *Ndn*- KO: 1.11 (0.75, 1.68), n = 4; MW, p=0.016. Scatter dot plots, report Q2 (Q1, Q3). (**E–H**) Real time and single living cell analyses of SERT uptake activity using the fluorescent substrate ASP+, a fluorescent substrate of SERT. (**E**) Kinetic experiment recordings of accumulation of ASP+ over time (5 min recording). Coefficient of Determination $R^2$: WT = 0.97; *Ndn* KO = 0.99; *Slc6a4*-KO = 0.93. Mean velocity (v) of ASP+ accumulation obtained by linear regression analyses of the slopes: (AUF.s$^{-1}$): WT: 0.36 ± 0.01,

*Figure 2 continued on next page*

*Figure 2 continued*

n = 18; *Ndn-KO*: 0.51 ± 0.01, n = 18, covariance (ANCOVA), p<0.0001. Non-specific accumulation of ASP+ fluorescence was evaluated in *Slc6a4*-KO neurons and found to be strongly low (0.02 ± 0.01 (n = 6 cells). (F) Saturation experiments using gradual concentration of APS+. Non-linear curve-fitting yielded a one-phase exponential association, with a Vmax (G) and Km (H): Vmax (AUF.s$^{-1}$): WT: 0.45 ± 0.05, n = 64; *Ndn*-KO: 0.84 ± 0.12, n = 67; *Slc6a4*-KO: 0.01 ± 0.03, n = 37, p<0.0001; Km (μM): WT: 6.03 ± 1.60, n = 64; *Ndn*-KO: 12.03 ± 3.55, n = 67; *Slc6a4*-KO: 1.83 ± 1.60, n = 37, p<0.0001. AUF: arbitrary unit of fluorescence. p-values determined by K-W test, followed by Dunn post-hoc test. Bar graphs represent mean ±SEM. *p<0.05; **p<0.01; ***p<0.001. Scatter dot plots, report Q2 (Q1, Q3). **p<0.01; ***p<0.001.

DOI: https://doi.org/10.7554/eLife.32640.007

The following figure supplements are available for figure 2:

**Figure supplement 1.** 5-HT metabolism, *Tph2* and *Slc6a4* transcripts quantification in *Ndn*-KO mice.

DOI: https://doi.org/10.7554/eLife.32640.008

**Figure supplement 2.** ASP+ uptake in neurons of raphe primary cultures.

DOI: https://doi.org/10.7554/eLife.32640.009

**Figure supplement 3.** Flow diagram of mice used for in vitro and in situ analyses in *Figures 1* and *2* and their corresponding supplement figures.

DOI: https://doi.org/10.7554/eLife.32640.010

Together, our data show that increased SERT expression in *Ndn*-KO mice underlies an increase of 5-HT reuptake, which accumulates in 5-HT LPAs. In the absence of any increase in 5-HT synthesis (and in fact increased 5-HT degradation), this sequence of events could be sufficient to cause a physiologically relevant decrease extracellular 5-HT.

## Genetic ablation or pharmacological inhibition of SERT uptake restores normal breathing in *Ndn*-KO mice

As exogenous 5-HT application stabilized respiratory rhythm of *Ndn*-KO mice in vitro, (*Zanella et al., 2008*), we hypothesized that SERT dysregulation observed in *Ndn*-KO mice might underlie their respiratory phenotype. To further investigate this causal link, we compared breathing parameters in WT, *Ndn*-KO, *Ndn/Slc6a4*-DKO and in *Ndn*-KO pups treated with Fluoxetine, a selective 5-HT reuptake inhibitor (SSRI) used clinically to increase extracellular 5-HT (*Figure 3A–B*). First, we confirmed that respiratory deficits, quantified as the percentage of mice exhibiting apnea (*Figure 3C*), the number of apneas per hour (*Figure 3D*), or the accumulated apnea duration (*Figure 3E*), were significantly increased in *Ndn*-KO compared to WT mice. These deficits were suppressed by reducing SERT function either by constitutive genetic inactivation (*Ndn/Slc6a4*-DKO pups) or by 10 days of Fluoxetine treatment (P5-P15; 10 mg/kg/day) in *Ndn*-KO pups (*Figure 3C–E*). Other basic respiratory parameters (minute ventilation, frequency of breathing, tidal volume) were unchanged between all genotypes (*Figure 3F–H*). Therefore, our results show that increasing extracellular 5-HT is sufficient to suppress apneas in juvenile *Ndn*-KO mice.

Since Fluoxetine treatment in early life has positive effects on apneas, we next questioned the long-term consequences of this treatment. Novel cohorts of WT, *Ndn*-KO and *Ndn*-KO pups were treated as above with Fluoxetine or vehicle and then submitted to plethysmography 0, 15 and 45 days after treatment (DAT) (*Figure 3—figure supplement 1A–B*). The positive effect of Fluoxetine on respiratory function in *Ndn*-KO pups at the end of treatment were confirmed in this cohort, but did not persist at 15 and 45 DAT (*Figure 3—figure supplement 1C–E*). Other respiratory parameters (minute ventilation, frequency of breathing, tidal volume) measured at 45 DAT were unchanged between all genotypes (*Figure 3—figure supplement 1F–H*).

An altered ventilatory response to hypercapnia was previously observed in adult *Ndn*-KO mice (*Zanella et al., 2008*), so we next investigated whether this deficit is apparent in P0-P1 pups. We examined the chemoreflex of *Ndn*-KO and WT neonates by initially subjecting them to a moderate hypercapnia (5 min; 4% $CO_2$) (*Figure 4A–C*). Under hypercapnic stress, WT but not *Ndn*-KO neonates progressively increased their respiratory frequency (Rf) (*Figure 4D*), leading to an increase in minute ventilation (volume breathed over 1 min,VE) (*Figure 4F*). In contrast, no significant effects of hypercapnia were detected on any respiratory variables in *Ndn*-KO pups and thus *Ndn*-KO pups appear relatively insensitive to hypercapnia. To determine whether altered central 5-HT transmission contributes to this effect, we performed electrophysiological recordings of rhythmic phrenic bursts using *en bloc* brainstem-spinal cord preparations from P0-P1 WT and *Ndn*-KO pups. During perfusion with physiological aCSF (pH 7.4), we found no significant difference in phrenic burst (PB) shape,

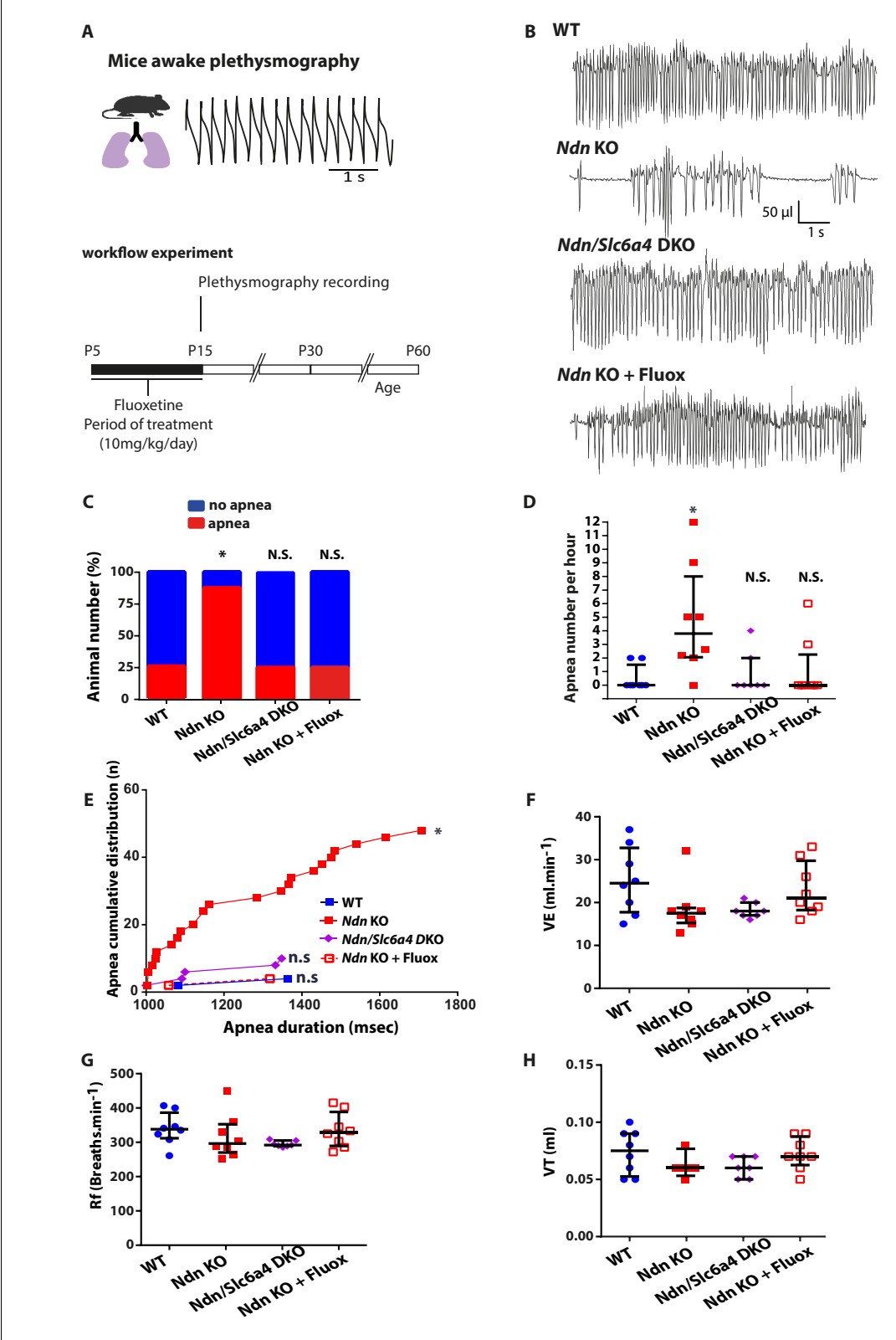

**Figure 3.** Genetic ablation or pharmacologic inhibition of SERT suppresses apnea and rescues central chemoreflex in *Ndn*-KO mice. (A) Workflow experiment of constant airflow whole body plethysmography performed in unanaesthetized, unrestrained WT, *Ndn-KO*, *Ndn/Slc6a4*-DKO and *Ndn-KO +Fluox* mice at the age of P15. *Ndn*-KO and WT animals (indicated here and in the figure as WT or *Ndn*-KO) have been pre-treated with 0.9% NaCl from age P5 to P15. *Ndn-KO* mice (indicated here and in the figure as *Ndn*-KO+Fluox) have been pre-treated with with Fluoxetine (10 mg/Kg/day) from
*Figure 3 continued on next page*

*Figure 3 continued*

age P5 to P15. (B) Plethysmographic recordings of WT, *Ndn*-KO, *Ndn/Slc6a4*-DKO and *Ndn*-KO+Fluox mice at the age of P15. (C–E) Quantification of apnea in P15 mice. (C) Proportion of apneic mice: WT: 2 of 8; *Ndn*-KO: 7 of 8; corresponding respectively to 25% and 87%; Chi$^2$ test, p=0.01. Genetic ablation of *Slc6a4* or early Fluoxetine treatment normalized the number of *Ndn-KO* apneic mice: *Ndn/Slc6a4* DKO: 2 of 8, 25%; Chi$^2$ test, p>0.99, N.S.; *Ndn*-KO+Fluox: 2 of 8, 25%; Chi$^2$ test, p>0.99, N.S. (D) Number of apnea in *Ndn-KO* compared to WT mice: WT: 0.0 (0.0, 1.5), n = 8; *Ndn*-KO: 3.8 (2.0, 8.0), n = 8; p=0.01. Genetic ablation of *Slc6a4* or Fluoxetine treatment normalized the number of apnea of *Ndn-KO* mice to WT values: *Ndn/Slc6a4*-DKO: 0.0 (0.0, 2.0), n = 8; p>0.99, N.S.; *Ndn*-KO+Fluox: 0.0 (0.0, 2.2), n = 8; p>0.99, N.S. p-values determined by KW test followed by Dunn post-hoc test with comparison to WT. (E) Cumulative distribution of apnea (number of cumulated values) over apnea duration (msec) in WT, *Ndn*-KO, *Ndn/Slc6a4*-DKO and *Ndn*-KO treated by Fluoxetine. Compared to WT, *Ndn*-KO mice demonstrated a significant increase of cumulative apnea both in term of number and duration (Kolmogorov-Smirnov test, p=0.01). However, such increase was normalized to WT after genetic deletion of *Slc6a4* or Fluoxetine treatment. (F–H) Basic breathing parameters: (F) Minute ventilation, VE (the total volume breathed over one min): WT: 24.5 (17.7, 32.7), n = 8; *Ndn*-KO: 17.5 (15.2, 18.7), n = 8; p=0.14, N.S.; *Ndn/Slc6a4*-DKO: 18.0 (17.0, 20.0), n = 8; p=0.25, N.S. and *Ndn*-KO+Fluox: 21.0 (18.2, 29.7), n = 8; p>0.99, N.S.. (G) Frequency of breathing, Rf (breaths/min): WT: 338 (312, 3867), n = 8; *Ndn*-KO: 296 (270, 352), n = 8; p=0.56, N.S.; *Ndn/Slc6a4*-DKO: 292 (2890, 305), n = 8; p=0.16, N.S. and *Ndn*-KO+Fluox: 329 (289, 388), n = 8; p>0.99, N.S. (H) Tidal Volume, VT (the volume flow per breath): WT: 0.07 (0.05, 0.09), n = 8; *Ndn*-KO: 0.06 (0.06, 0.06), n = 8; p=0.38, N.S.; *Ndn/Slc6a4*-DKO: 0.06 (0.05, 0.07), n = 8; p=0.51, N.S. and *Ndn*-KO+Fluox: 0.07 (0.06, 0.08), n = 8; p>0.99, N.S. p-values determined by K-W test followed by Dunn post-hoc test with comparison to WT. Scatter dots represent Q2 (Q1, Q3). N.S.: non-significant; *p<0.05.

DOI: https://doi.org/10.7554/eLife.32640.011

The following figure supplements are available for figure 3:

**Figure supplement 1.** Early life Fluoxetine treatment has only short-term positive effects on *Ndn*-KO apneas.

DOI: https://doi.org/10.7554/eLife.32640.012

**Figure supplement 2.** Early life treatment of Fluoxetine on respiratory apnea in wild-type mice.

DOI: https://doi.org/10.7554/eLife.32640.013

amplitude or discharge frequency (PBf) between WT and *Ndn*-KO pups (*Figure 4G–H*). As expected, PBf in WT preparations progressively increased upon acidosis (pH = 7.1, *Figure 4I,L*). However, this effect was not observed in *Ndn*-KO preparations (*Figure 4J,L*).

We then assessed whether increasing extracellular 5-HT could rescue chemoreflex sensitivity in this preparation. Bath application of Fluoxetine (20 μM) prior to acidosis did not affect baseline PBf of *Ndn*-KO preparations (*Figure 4K,L*), but instead significantly increased PBf responses to acidosis to levels indistinguishable from WT controls (*Figure 4K,L*). Qualitatively similar responses were observed in experiments in which a 5-HT1A receptor agonist (8OHDPAT) was substituted for Fluoxetine (*Figure 4—figure supplement 1A–D*). We therefore conclude that the central chemoreceptor hyposensitivity characteristic of the *Ndn*-KO model can be restored by pharmacological manipulations that increase extracellular 5-HT and/or stimulate 5-HT1A-R activity.

## Early life Fluoxetine-treatment has deleterious long-term respiratory consequences in WT mice

Although Fluoxetine had beneficial but transient effects on apnea incidence in *Ndn*-KO mice, we observed deleterious and long-lasting effects on respiratory function in WT controls. Early life Fluoxetine-treatment induced a significant increase in the number of apneic mice, the frequency of apneas, and the cumulative distribution of apneas at all timepoints measured (0, 15 and 45 DAT, *Figure 3—figure supplement 2A–E*), such that measurements at 45 DAT in WT mice (*Figure 3—figure supplement 2*) were similar to those obtained in *Ndn*-KO mice (*Figure 3—figure supplement 1*). The sensitivity of WT brainstem-spinal cord preparations, treated with Fluoxetine or with 8OHDPAT, to acute acidosis was similarly affected (*Figure 4—figure supplement 2A–D*). In neutral aCSF, neither Fluoxetine (*Figure 4—figure supplement 2C*) or 8OHDPAT (*Figure 4—figure supplement 2D*) affected resting PBf of WT *en bloc* preparations but instead abolished the normal increases in PBf responses to acidosis. Thus, we confirm that Fluoxetine treatment abolishes the capacity of WT mice to respond to acidosis (*Voituron et al., 2010*), and we propose a role for 5-HT1A-R activity in this response. We show here, for the first time, adverse effects of Fluoxetine on breathing outcomes.

## Conclusion

Previously, a pleiotropic function of Necdin has been reported in different neuronal populations and at different developmental stages. Concerning the 5-HT system, an expression of Necdin was observed in virtually all 5-HT neurons (*Zanella et al., 2008*) and an alteration of the 5-HT system in

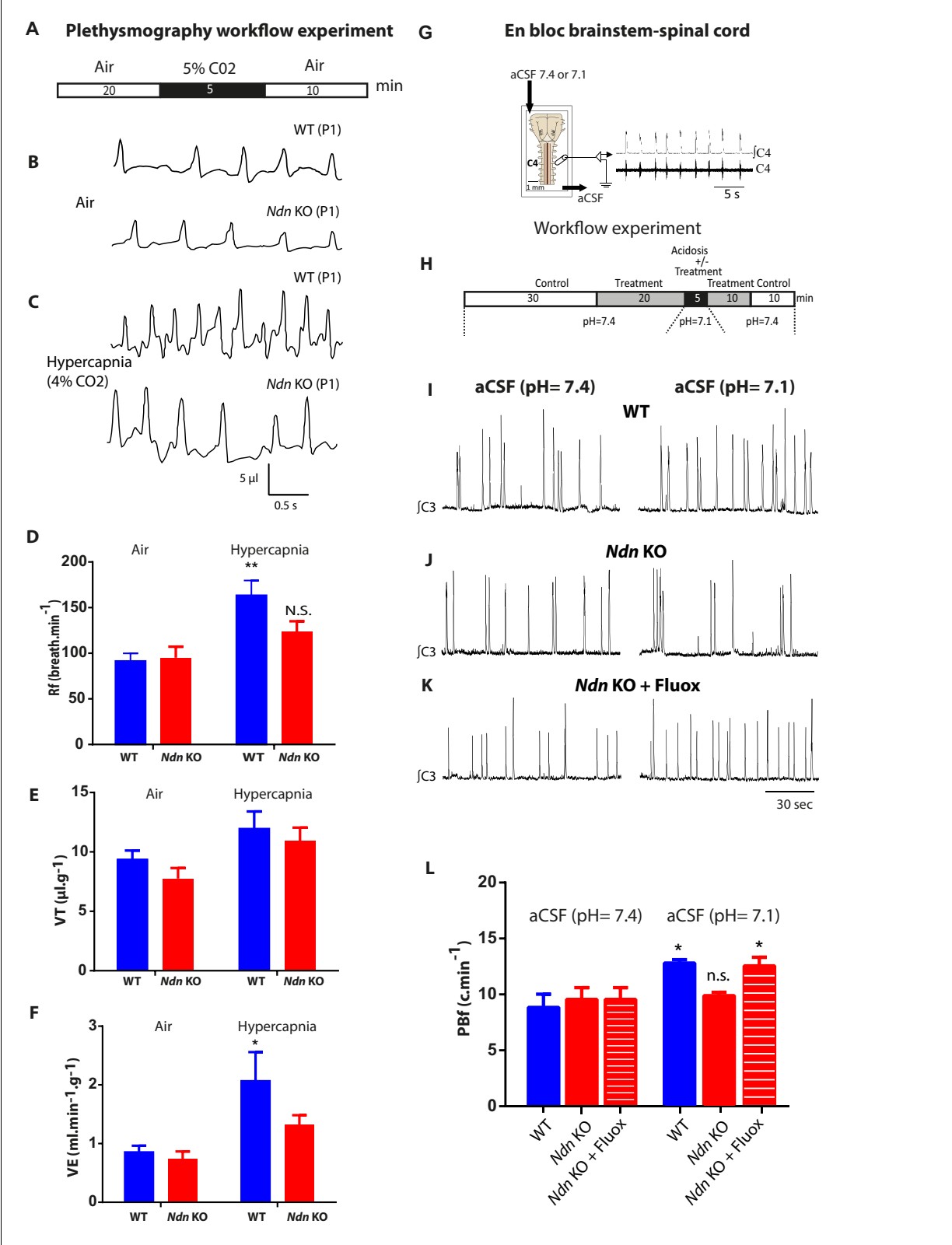

**Figure 4.** Alteration of respiratory chemoreflex in *Ndn*-KO neonates is rescued by Fluoxetine. (A–F) Effect of hypercapnia on in vivo ventilatory parameters of WT and *Ndn*-KO neonates. (A) Workflow experiment of constant airflow whole body plethysmography performed in unanaesthetized, unrestrained WT, *Ndn-KO* neonates at P0-P1 when breathing either air or hypercapnic mixture containing 4% $CO_2$ in air for 5 min. Data for analyses were collected in the last 5 min (air) or the last min (hypercapnia). (B–C) (B) Plethysmographic recordings of WT and *Ndn*-KO neonates when breathing

*Figure 4 continued on next page*

*Figure 4 continued*

air or (C) at 5th min upon hypercapnic respiratory challenge. (D) Respiratory frequency (Rf) in WT and *Ndn*-KO pups when subjected to hypercapnic stress: WT Air: 91 ± 8; WT hypercapnia: 163 ± 16; n = 8, p=0.004; *Ndn*-KO Air: 95 ± 12; *Ndn*-KO hypercapnia: 123 ± 11; n = 8, p=0.31, N.S. p-values determined by two-way-ANOVA test followed by Bonferroni post-hoc test. Bar graphs represent mean ±SEM; **p<0.01; N.S.: non-significant. (E) Tidal Volume (VT) ($\mu$l.g$^{-1}$) in WT neonates: WT Air: 9 ± 0.7; WT hypercapnia: 12 ± 1; n = 8, p=0.12 N.S.; in *Ndn*-KO: *Ndn*-KO Air: 7.5 ± 0.9; *Ndn*-KO hypercapnia: 10.9 ± 1.1; n = 8, p=0.055. p-values determined by two-way ANOVA test followed by Bonferroni post-hoc test. (F) Minute Ventilation (VE) (ml.min$^{-1}$.g$^{-1}$) in WT neonates: WT Air: 0.8 ± 0.1; WT hypercapnia: 2.1 ± 0.1; n = 8, p=0.01; in *Ndn*-KO: *Ndn*-KO Air: 0.7 ± 0.1; *Ndn*-KO hypercapnia: 1.3 ± 0.2; n = 8, p=0.3 N.S. p-values determined by two-way ANOVA test followed by Bonferroni post-hoc test. Bar graphs represent mean ±SEM. *p<0.05. (G–H) Effect of Fluoxetine treatment on the resting phrenic burst frequency (PBf) and the PBf response to acidosis in *Ndn*-KO medulla preparations. (G) Electrophysiological recordings of PBf produced in vitro in WT and *Ndn*-KO *en bloc* brainstem-spinal cord preparations at P0-P1 when superfused first with neutral artificial cerebrospinal fluid (aCSF) (pH 7.4) and then acidified aCSF (pH 7.1). (H) Workflow experiment of the electrophysiological recordings on medullary preparations to assess central chemosensitivity in WT and treated with Fluoxetine (20 $\mu$M) or untreated *Ndn*-KO mice. (I–K) Examples of continuous electrophysiological recordings of rhythmic phrenic bursts produced in *en bloc* brainstem-spinal cord preparations of (I) one WT, (J) one *Ndn*-KO and (K) one *Ndn*-KO treated with Fluoxetine (20 $\mu$M) pup and superfused with first neutral aCSF (pH 7.4) (left column recordings) or acidified aCSF (pH 7.1) (right column recordings). (L) Quantifications of the resting PBf (c.min$^{-1}$) of *en bloc* preparations superfused with neutral aCSF (pH 7.4) or acidified aCSF (pH 7.1) respectively of WT (WT (pH 7.4): 8.8 c.min$^{-1}$ ±1.2; WT (pH 7.1) 12.8 ± 0.3 c.min$^{-1}$; n = 12, p<0.001), *Ndn*-KO (*Ndn*-KO (pH 7.4): 9.5 ± 1.0; *Ndn*-KO (pH 7.1): 9.8 ± 0.3 c.min$^{-1}$; n = 12, p=0.41, N.S.) and *Ndn*-KO treated with Fluoxetine: (*Ndn*-KO+Fluox (pH 7.4): 9.6 ± 0.3; *Ndn*-KO+Fluox (pH 7.1): 12.6 ± 0.7; n = 12, p=0.04). Noticeably, under neutral aCSF (pH 7.4) no difference was observed between WT, *Ndn*-KO and *Ndn*-KO+Fluox. However, in acidified aCSF (pH 7.1), Fluoxetine significantly increased the PBf of *Ndn*-KO preparations. p-values determined by two-way ANOVA test followed by Tukey post-hoc test. Bar graphs represent mean ±SEM. N.S: non-significant; *p<0.05.

DOI: https://doi.org/10.7554/eLife.32640.014

The following source data and figure supplements are available for figure 4:

**Source data 1.** Plethysmography data before and after hypercapnia in WT and *Ndn*-KO mice.
DOI: https://doi.org/10.7554/eLife.32640.018

**Source data 2.** Electrophysiology data of rhythmic phrenic bursts frequency during acidosis in WT and *Ndn*-KO preparations - before and after Fluoxetine treatment.
DOI: https://doi.org/10.7554/eLife.32640.019

**Source data 3.** Relates to *Figure 4—figure supplements 1* and *2*.
DOI: https://doi.org/10.7554/eLife.32640.020

**Figure supplement 1.** Effect of pre-treatment with the 5-HT1A-R agonist 8OHDPAT on the resting PBf and the PBf response to acidosis in *Ndn*-KO *en bloc* brainstem-spinal cord preparations of P0-P1 pups.
DOI: https://doi.org/10.7554/eLife.32640.015

**Figure supplement 2.** Effects of Fluoxetine and of the 5-HT1A-R agonist 8OHDPAT on the resting PBf and the PBf response to acidosis in wild-type medulla preparations.
DOI: https://doi.org/10.7554/eLife.32640.016

**Figure supplement 3.** Flow diagram of mice used for ex vivo and in vivo analyses in *Figures 3* and *4* and their corresponding supplement figures.
DOI: https://doi.org/10.7554/eLife.32640.017

embryonic and postnatal development was partially described in both *Ndn*-KO (*Ndn*<sup>tm1-Stw</sup>and *Ndn*<sup>tm1-Mus</sup>) mouse models, with alterations in 5-HT axonal bundle projections (*Lee et al., 2005*; *Pagliardini et al., 2005*) and 5-HT fibers containing swollen 5-HT 'varicosities' (*Pagliardini et al., 2005*; *Zanella et al., 2008*). Furthermore, an alteration of 5-HT metabolism (*Zanella et al., 2008*) was observed in mutant neonates suggesting that it might alter 5-HT modulation of the Respiratory Rhythm Generator. Finally, an in vitro exogenous application of 5-HT on brainstem-spinal cord preparations of *Ndn* mutant mice alleviates the incidence of apneas (*Pagliardini et al., 2005*; *Zanella et al., 2008*). Despite those observations, the pathological mechanism responsible for the serotonopathy in *Ndn*-KO mice and the causal link between this serotonopathy and the breathing alterations were not investigated. Here, we aimed to answer those questions.

Noticeably, all previous studies have been performed on heterozygous *Ndn*-deficient mice, with a deletion of the *Ndn* paternal allele only (*Ndn*+m/-p), the maternal allele being normally silent. However, we have shown that, due to a faint and variable expression of the *Ndn* maternal allele (+m), *Ndn*+m/-p mice present a variability in the severity of respiratory phenotype compared with the *Ndn*-/- mice (here named *Ndn*-KO) (*Rieusset et al., 2013*). For instance, reduction of 5-HT neurons was not previously found significant in the *Ndn* + m/-p mice (*Zanella et al., 2008*) but has been found significantly reduced in the *Ndn*-/- mice. In order to avoid such variability and to get consistent results, we chose here to study *Ndn*-/- mice.

Here, we have shown that Necdin plays a pleiotropic role in the development of 5-HT neuronal precursors that guides the development of central serotonergic circuits and the physiological activity of mature 5-HT neurons. Our results suggest that Necdin controls the level of SERT expression in 5-HT neurons and that lack of Necdin increases the quantity and activity of SERT leading to an increased reuptake and intra-cellular accumulation of 5-HT, as visualized by 5-HT LPAs, leading to a reduction in available extracellular 5-HT. Importantly, in vivo inhibition of SERT activity, genetically or pharmacologically (Fluoxetine treatment), is sufficient to prevent the formation of those 5-HT LPAs and suppresses the apnea observed in *Ndn*-KO mice. We also demonstrate, using an ex vivo approach, that the altered chemosensitivity to $CO_2$/acidosis is caused by a central 5-HT deficit and is rescued by Fluoxetine-treatment. We conclude that an increase of 5-HT reuptake is the main cause of breathing deficits (central apnea and hypercapnia response) in *Ndn*-KO mice.

Unexpectedly, we reveal an adverse and long-term effect of early life administration of Fluoxetine on the breathing (apneas, chemosensitivity to $CO_2$/acidosis) of healthy mice. Previous adverse effects have been observed on anxiety and depression (*Glover and Clinton, 2016*; *Millard et al., 2017*) after an early postnatal administration of Fluoxetine but the respiratory deficits are reported here for the first time and should be further investigated in another study.

Respiratory failure in patients with PWS constitute a challenging issue since it is the most common cause of death for 73% of infants and 49% of children, (*Butler et al., 2017*). Death is often linked to respiratory infection or respiratory disorder and may be sudden, with some reported cases of sudden death occurring at night (*Gillett and Perez, 2016*). In PWS patients, any environmental acute respiratory challenge caused by, for instance, a respiratory tract infection, high altitude or intense physical activity further exacerbates their inherent disability (blunted response to hypoxima/hypercapnia) to adapt an respiratory response. Until now, the underlying pathology for respiratory failure remained elusive and did not appear to be impacted by recent advancements in treatment modalities (*Butler et al., 2017*). Although oxygen treatment is efficient in preventing the hypoxemia induced by central apneas (*Urquhart et al., 2013*), such treatment is physically constraining. Within the context of PWS, the current study points towards a critical link between Necdin, serotonopathy, and chemosensing, a function in which brainstem serotonergic circuits play a critical role. Since our study shows that Fluoxetine can suppress apnea and restore chemosensitivity, we propose that Fluoxetine might be an appropriate 'acute' treatment that could be considered for Prader-Willi infants/children when they present the first signs of any breathing difficulties.

## Materials and methods

### Animals

Mice were handled and cared for in accordance with the Guide for the Care and Use of Laboratory Animals (N.R.C., 1996) and the European Communities Council Directive of September 22th 2010 (2010/63/EU, 74). Experimental protocols were approved by the institutional Ethical Committee guidelines for animal research with the accreditation no. B13-055-19 from the French Ministry of Agriculture. All efforts were made to minimize the number of animals used. Necdin is an imprinted gene, paternally expressed only (*Figure 2—figure supplement 3* and *Figure 4—figure supplement 3*). In order to avoid a variability in our results due to a stochastic and faint expression of the maternal allele (*Rieusset et al., 2013*), we worked with the *Ndn*$^{tm1-Mus}$ strain and decided to study *Ndn*-/- mice (named here *Ndn*-KO), instead of *Ndn*+m/-p mice as it has been done previously.

Fluoxetine was obtained from Sigma (Saint-Quentin Fallavier, France) for cell culture and *en bloc* medullary experiments and from Mylan pharma for in vivo experiments.

### Transgenic mice

We bred *ePet-EYFP*-expressing (*Scott et al., 2005a*; *Scott et al., 2005b*) or *Slc6a4-Cre* Knock-in (*Zhuang et al., 2005*) mice with *Ndn*-KO (*Muscatelli et al., 2000*) mice, all on C57BL/6 background. Protocols of genotyping mice have been previously described for *Pet-EYFP* (*Hawthorne et al., 2010*), *Ndn*-KO (*Rieusset et al., 2013*) and *Sert*-Cre Knock-in mice (*Zhuang et al., 2005*), in which the *Slc6a4* gene was replaced by Cre was referred to in the text as *Slc6a4*-KO. Breeding of *Slc6a4*-KO with *Ndn*-KO mice was referred to in the text as *Ndn-Slc6a4*-DKO.

## Immunohistochemistry and quantification

Tissue preparation and IHC were performed as previously described (*Rieusset et al., 2013*). Antibodies used were: rabbit polyclonal anti-Necdin (07–565; Millipore, Bedford, MA, USA; 1:500), mouse monoclonal anti-GFP (Interchim, NB600-597; 1:500), goat polyclonal anti-5HT (Immunostar, 20079; 1:300). Sections were examined on a Zeiss Axioplan two microscope with an Apotome module.

Brainstem structures were sampled by selecting the raphe obscurus area and counting was performed on three sagittal sections/animal of 100 μm which represent the entire PET1-YFP positive cell population of the raphe obscurus (ROb/B2) and pallidus (RPa/B1), both nuclei being difficult to separate. For each section, a Z-stack composed of 10 confocal images (8 μm focal spacing) was acquired. For quantification, stereological method has been applied on each Z-stack image using the eCELLence software developed by Glance Vision Technologies (Italy). The total cell number/per animal was obtained by summing the sub-total of cells counted for the 3 Z-stacks.

Images of 5-HT LPAs were acquired using a confocal microscope (Olympus). Between 4 and 8 fibers/brain region for each animal (3WT and 3 KO) were analyzed for the presence of 5-HT LPAs (>1.8 μm$^2$) on 100 μm long fiber. The size of 5-HT LPAs was quantified using Image J. 5-HT LPA diameter has been defined *ad arbitrium* as the size of the largest 5-HT punctiform labelling found in the WT fibers.

## Organotypic slice cultures and time lapse experiments

Slice cultures from E11.5 embryonic mouse brainstems were prepared from *Pet-EYFP* and *Ndn KO/ Pet-EYFP* mice. Thick coronal sections (250 μm) brainstem were cut using a tissue chopper and cultured in Neurobasal medium (Thermofisher) containing 2% B27 (Thermofisher), 4% horse serum, 10 μg/ml insulin, 200 mM HEPES, 1% Antibiotic Antimycotic (Thermofisher). For time lapse experiments, the dishes were mounted in a $CO_2$ incubation chamber (5% $CO_2$ at 37°C) fitted onto an inverted confocal microscope (LSM510, Zeiss). Acquisitions of the region containing raphe Pet-EYFP +neurons were performed every 10 min for up to 15 hr. Cell coordinates, velocity, and tortuosity (total length of the track/direct distance from the first to the last point) were calculated using MtrackJ plugin of Image J.

## Electrophysiology patch-clamp

Sagittal slices that included the raphe (400 μm thick) were cut from brainstems of 2 week old *Pet-EYFP* and *Ndn-KO/Pet-EYFP* mice. Whole-cell recordings were made from YFP+ cells in the region of the B4 raphe nucleus. During recordings, slices were continuously perfused with artificial cerebrospinal-fluid (aCSF) at 37°C. Patch pipettes (4–5 MΩ) were filled with an internal solution with the following composition (in mM): 120 KGlu, 10 KCl, 10 $Na_2$-phosphocreatine, 10 HEPES, 1 $MgCl_2$, 1 EGTA, 2 ATP Na, 0.25 GTP Na; pH = 7.3 adjusted with KOH. Current clamp at i = 0 were recorded with a HEKA amplifier and acquired using PatchMaster software (HEKA). Offline analysis was performed with Clamfit 10.3.

## In vitro recordings from *en bloc* brainstem-spinal cord preparations

As previously reported (*Berner et al., 2012*), the medulla and cervical cord of P0-P1 neonatal mice were dissected, placed in a 2 ml in vitro recording chamber, bubbled with carbogen, maintained at 27°C and superfused (3.5–4.5 ml per min) with aCSFcomposed with (mM): 129.0 NaCl, 3.35 KCl, 21 $NaHCO_3$, 1.26 $CaCl_2$, 1.15 $MgCl_2$, 0.58 $NaH_2PO_4$, and 30.0 D-glucose ('Normal aCSF': pH 7.4) or using the same components except with 10 mM $NaHCO_3$ ('Acidified aCSF': pH 7.1). Inspiratory discharges of respiratory motoneurons were monitored by extracellular recording with glass suction electrodes applied to the proximal cut end of C4 and C3 spinal nerves roots. Axoscope software and Digidata 1320A interface (Axon Instruments, Foster, CA, USA) were used to collect electrophysiological data. Offline analysis was performed with Spike 2 (Cambridge Electronic Design, UK) and Origin 6.0 (Microcal Software, Northampton, MA, USA) software for PC. Burst frequency was analyzed and calculated as the number of C4 bursts per minute. The values of inspiratory burst frequency were calculated as the mean of the last 3 min of any condition: ACSF (7.4) and ACSF (7.1). Standardized experiments in WT and *Ndn*-KO preparations were repeated on different preparations

from different litters. For a given preparation, only one drug was applied and only one trial was performed.

## RT-qPCR

For RT-qPCR, mice were sacrificed at P1, the brainstem dissected, and tissues were rapidly collected and frozen in liquid nitrogen prior to RNA isolation using standard conditions. RNA, reverse transcription and real time PCR were conducted as previously described (*Rieusset et al., 2013*). Sequences of the various primer pairs used for qPCR, as well as the slope of the calibration curve established from 10 to $1 \times 10^9$ copies and qPCR efficiency E, were as follow: *Tph2*: F: 5'-GAGC TTGATGCCGACCAT-3'; R: 5'-TGGCCACATCCACAAAATAC-3'; *Slc6a4*: F:5'-CATATGCTACCAGAA TGGTGG-3'; R:5'-AAGATGGCCATGATGGTGTAA-3'. For each sample, the number of cDNA copies was normalized according to relative efficiency of RT determined by the standard cDNA quantification. Finally, gene expression was expressed as the cDNA copy number quantified in 5 µL aliquots of RT product.

## Western blot

Newborn mice were sacrificed and brainstems were immediately dissected and snap-frozen in liquid nitrogen and stored at −80°C until protein extraction. Protein extraction was conducted as previously described (*Felix et al., 2012*). Membranes were blocked with PBS containing 5% BSA for 1 hr, followed by an overnight incubation at 4°C with the following primary antibodies: guinea pig anti-SERT (1/2000, Frontier Institute), mouse anti-B3 tubulin (1/2000, ThermoFisher Scientific). Membranes were then washed and incubated 2 hr with either anti-guinea pig (1/1000, ThermoFisher Scientific), or anti-mouse (1/2000; DAKO) horseradish peroxidase-conjugated secondary antibodies. Visualisation was performed using the Super signal West-pico chemolumniscent substrate (Pierce, Thermo Scientific, France). Quantification was performed using ImageJ.

## Biochemical analysis of the medullary serotonergic system

Pregnant mice were killed by cervical dislocation at gestational day E18.5 and fetuses were removed, decapitated, and the medulla dissected and stored at −80°C until measurements. Medullary 5-HT, its precursor L-tryptophan (L-Trp), and its main metabolite, 5-hydroxy-indol acid acetic (5-HIAA), were measured with high-pressure liquid chromatography separation and electrochemical detection (Waters System: pump P510, electrochemical detector EC2465; Atlantis column DC18; mobile phase: citric acid, 50 mM; orthophosphoric acid, 50 mM; sodium octane sulfonic acid, 0.112 mM; EDTA, 0.06 mM; methanol, 5%; NaCl, 2 mM; pH 2.95). Contents are expressed in nanograms per medulla.

## Raphe primary neuronal culture and live cell uptake assay

### Raphe primary cell culture

Newborn mice (n = 6 per culture) were decapitated, brainstems extracted, the meninges removed and the medial part of the brainstem dissected. Tissues were enzymatically digested at 37°C for 30 min with HBSS containing 2 mg/mL of filter-sterilized papain. Cells were resuspended in Neurobasal medium (Thermofisher) containing 2% B27 (Thermofisher), 0.5 mM L-glutamine, glucose (50 mM), 50 ng/ml NGF, 10 ng/ml bFGF, 10 µg/ml insulin. $2 \times 10^5$ cells were plated on round 14 mm glass coverslip pre-coated with Polyethyleneimine (20 µg/ml). Cells were cultured during 8 days in presence of 5% of NU serum (Becton Dickinson) during the first 2 days. Immunocytochemistry was performed to verify presence of 5-HT$^+$ neurons in the culture.

## Live cell imaging of (4-(4-(dimethylamino)styryl)-N-methylpyridinium (ASP+) uptake

Cells were placed in a bath chamber on the stage of an inverted microscope (Nikon eclipse TE300) and perfused (2 ml/min) with Krebs medium (mM): 150 NaCl; 2.5 KCl; 2 CaCl$_2$; 2 MgCl$_2$; 2.5 Hepes acide; 2.5 Hepes-Na; pH 7,4. Time-lapse cell acquisition was started when ASP+ (1, 2, 5, 10, 15 or 20 µM) was added to the perfusion. ASP+ was excited at 488 nm and fluorescence was captured at 607 nm every 10 s for 5 min using Metamorph software (MolecularDevices). Each ASP+ concentration was tested on three different cultures for WT and *Ndn*-KO and one for *Ndn/Slc6a4*-DKO. Cells

placed on the coverslip were replaced for each concentration tested. For each ASP+ cells, an ROI of the same surface was delineated on the soma in order to measure pixel intensity in arbitrary fluorescence units. 6 ROI were determined at each measurement. Data were background subtracted and ASP+ fluorescence intensity was expressed as a function of initial fluorescence intensity.

## In vivo recordings of breathing parameters by plethysmography

Breathing of unrestrained, non-anesthetized mice was recorded using constant air flow whole-body plethysmography filled with air or 4% $CO_2$ in air (EMKA Technologies, Paris, France). Neonatal mice (P0-P1) were recorded in 25 ml chambers (calibrated by injecting 50 µl of air) maintained at neonatal thermoneutral ambient temperature ($32 \pm 0.5$°C). For adolescent and adult mice (P15-P30-P60), four plethysmography 200 ml chambers containing air or (calibrated by injecting 1 ml of air) maintained at $25 \pm 0.5$°C were used to allow simultaneous measurements. Analog signals were obtained using an usbAMP device equipped with four inputs and processed using EMKA technologies IOX software (EMKA Technologies, Paris, France). For neonatal mice, we measured mean respiratory frequency (Rf, expressed in cycles per minute) during quiet periods when mice breathed air or 5 min after breathing hypercapnic air. For adolescent and adult mice respiratory parameters (frequency, tidal volume, minute ventilation) were recorded over 30 min after an initial 30 min period of stabilization in the apparatus.,Apnea was defined as a prolonged expiratory time (four times eupneic *expiratory time*), which corresponds to a threshold of 1 s.

## Statistical analysis

Analyses were performed using two-tailed non-parametric statistical tools due to the size of the samples (GraphPad, Prism software). Values are indicated as following: (Q2 (Q1, Q3), n; statistical test, p-value) where Q2 is the median, Q1 is the first quartile and Q3 is the third quartile and scatter dot plots report Q2 (Q1, Q3). Histograms report the mean ±SEM. The level of significance was set at a p-value less than 0.05. Appropriate tests were conducted depending on the experiment and are indicated in the figure legends. Mann-Whitney (MW) test was performed to compare two unmatched groups: differences between WT and *Ndn*-KO (*Figure 1* and *Figure 2—figure supplement 1*). Kolmogorov-Smirnov test was performed to compare the cumulative distribution of two unmatched groups: differences between WT and *Ndn*-KO in apnea accumulation over time (*Figure 3E*; *Figure 3—figure supplement 1E*; *Figure 3—figure supplement 2E*). Chi-square test was performed to compare two groups of animal (WT and *Ndn*-KO) with categorical outcome variable (apnea or no apnea) (*Figure 3C*; *Figure 3—figure supplement 2C*). Kruskal-Wallis (KW) followed by a post hoc test Dunn test was performed to compare three or more independent groups (*Figure 2G,H*; *Figure 3D,F–H*); Friedman test followed by a post hoc test Dunn test was performed to compare matched groups (*Figure 4—figure supplement 2C,D*). Two-way ANOVA followed by Bonferroni post-hoc test was performed to compare two factors (*Figure 2B*). Two-way repeated-measure (RM) ANOVA was performed to compare two factors (genotype compared either to time, drug treatment or respiratory challenge) with repeated measure matched by time or respiratory challenge (*Figure 3—figure supplement 1D*; *Figure 3—figure supplement 2D*;*Figure 4D–F,L* and *Figure 4—figure supplement 1D*); genotype and respiratory challenge. ANCOVA was performed to compare slopes of two regression lines (WT *versus Ndn*-KO: *Figure 2E*). *p<0.05; **p<0.01; ***p<0.001; ****p<0.0001.

## Acknowledgements

We thank Camille Dumon and Magdalena Assael for their technical help and the members of the animal facility, genotyping and imaging platforms of INMED laboratory. We thank Pr Keith Dudley for comments and careful reading of the manuscript. Opinions expressed are those of the authors exclusively. This study has been supported by INSERM, CNRS and ANR (Prageder N° ANR-14-CE13-0025-01) grants. YS was supported by Stiftelsen Frimurare Barnhuset i Stockholm grants and N.K.H. Kronprinsessan Lovisas Forening for barnasjukvard; AB was supported by FPWR fellowship grant.

# Additional information

## Funding

| Funder | Grant reference number | Author |
|---|---|---|
| Institut National de la Santé et de la Recherche Médicale | | Valéry Matarazzo<br>Laura Caccialupi<br>Fabienne Schaller<br>Nazim Kourdougli<br>Alessandra Bertoni<br>Clément Menuet<br>Patricia Gaspar<br>Françoise Muscatelli |
| Centre National de la Recherche Scientifique | | Laurent Bezin<br>Pascale Durbec<br>Gérard Hilaire<br>Françoise Muscatelli |
| Agence Nationale de la Recherche | PRAGEDER ANR14-CE13-0025-01 | Valéry Matarazzo<br>Fabienne Schaller<br>Yuri Shvarev<br>Clément Menuet<br>Nicolas Voituron<br>Gérard Hilaire<br>Françoise Muscatelli |
| Stiftelsen Frimurare Barnhuset i Stockholm | | Yuri Shvarev |
| Kronprinsessan Lovisas Forening for Barnasjukvard | | Yuri Shvarev |

The funders had no role in study design, data collection and interpretation, or the decision to submit the work for publication.

## Author contributions

Valéry Matarazzo, Conceptualization, Supervision, Investigation, Writing—original draft, Project administration, Writing—review and editing; Laura Caccialupi, Conceptualization, Formal analysis, Supervision, Validation, Investigation, Visualization, Methodology, Writing—original draft, Writing—review and editing; Fabienne Schaller, Yuri Shvarev, Formal analysis, Validation, Investigation, Visualization, Methodology; Nazim Kourdougli, Alessandra Bertoni, Formal analysis, Investigation, Visualization, Methodology; Clément Menuet, Nicolas Voituron, Investigation, Visualization, Methodology, Writing—review and editing; Evan Deneris, Resources, Investigation, Visualization, Methodology; Patricia Gaspar, Resources, Methodology, Writing—review and editing; Laurent Bezin, Methodology, Writing—review and editing; Pascale Durbec, Visualization, Methodology; Gérard Hilaire, Conceptualization, Validation, Visualization; Françoise Muscatelli, Conceptualization, Data curation, Formal analysis, Supervision, Funding acquisition, Validation, Investigation, Visualization, Writing—original draft, Project administration, Writing—review and editing

## Author ORCIDs

Valéry Matarazzo (iD) https://orcid.org/0000-0002-0833-203X
Yuri Shvarev (iD) http://orcid.org/0000-0001-6622-1453
Nazim Kourdougli (iD) http://orcid.org/0000-0002-8725-792X
Clément Menuet (iD) http://orcid.org/0000-0002-7419-6427
Nicolas Voituron (iD) http://orcid.org/0000-0002-2092-4900
Patricia Gaspar (iD) http://orcid.org/0000-0003-4217-5717
Pascale Durbec (iD) http://orcid.org/0000-0002-9660-1809
Françoise Muscatelli (iD) https://orcid.org/0000-0003-4001-6528

## Ethics

Animal experimentation: Mice were handled and cared for in accordance with the Guide for the Care and Use of Laboratory Animals (N.R.C., 1996) and the European Communities Council Directive

of September 22th 2010 (2010/63/EU, 74). Experimental protocols were approved by the Institutional Ethical Committee guidelines for animal research with the accreditation no. B13-055-19 from the French Ministry of Agriculture.

## Decision letter and Author response
Decision letter https://doi.org/10.7554/eLife.32640.023
Author response https://doi.org/10.7554/eLife.32640.024

## Additional files

### Supplementary files
• Transparent reporting form
DOI: https://doi.org/10.7554/eLife.32640.021

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
