## [Decision Letter]

Thank you for submitting your article "Necdin shapes serotonergic development and SERT activity modulating breathing in a mouse model for Prader-Willi Syndrome" for consideration by *eLife*. Your article has been reviewed by two peer reviewers, and the evaluation has been overseen by Joseph Gleeson as Reviewing Editor and a Senior Editor. The reviewers have opted to remain anonymous.

The reviewers have discussed the reviews with one another and the Reviewing Editor has drafted this decision to help you prepare a revised submission.

The reviewers and editors appreciated that the work clearly demonstrated the respiratory defect in mouse pups lacking necdin is an increase in SERT activity, and that inhibiting SERT could be a therapeutic approach for Prader Willi patients. Overall this is an interesting study with therapeutic possibilities for Prader-Willi and other disorders featuring respiratory suppression with insensitivity to hypoxia/hypercapnia.

Summary:

Matarazzo and colleagues investigate the respiratory system in mice deficient for necdin, as a model for neonatal respiratory deficits present in children with Prader Willi Syndrome. They report that lack of necdin disturbs the migration of serotonin neuronal precursors and the morphology of serotonergic neurons. In these mice, either deletion of the serotonin transporter (Sert) or treatment with fluoxetine, a SERT reuptake inhibitor, restores normal breathing. They propose that the respiratory defect in mouse pups lacking necdin is an increase in SERT activity, and that inhibiting SERT could be a therapeutic approach for Prader Willi patients.

In general, this is a well designed, implemented, and written body of work that develops insight into the development of breathing abnormalities in Prader-Willi and other disorders featuring respiratory suppression with insensitivity to hypoxia/hypercapnia and suggests potential treatment for these problems.

Essential revisions:

1) The histological analysis does not indicate enough specifics on how quantification was performed, notably it does not appear that stereological methods were utilized. In the absence of such methods there are significant concerns for bias.

2) The authors focus on the serotonin system and the effect of modulating the 5-HT system genetically or pharmacologically. Previous studies have shown that other excitatory neuromodulators (Substance P and TRH) likewise partially rescue the respiratory deficit in brainstem-spinal cord preparations from Ndn-KO mouse pups (Pagliardini Advances in Experimental Medicine and Biology 2008). This raises the possibility that the serotonergic deficit is only one of many deficits in this genetic mutant. This point should be addressed and weighed in light of previous publications.

3) The study would have much higher impact if the truly novel findings were disentangled from replication of previous results, although this replication is also important to report. The authors fail to place their experiments in the context of previously published work, in some cases contradicting their own previous studies (e.g. number of 5_HT neurons, (Zanella, 2008)) but confirming that of others (Pagliardini, 2005) with no reference to either study. This will put undue strain on readers to identify which are new observations and which are repeats of previously published observations.

4) The abnormality of serotonergic function in the respiratory system in the hindbrain of Ndn-null mice has been studied by several groups over the last decade, compromising the novelty of the study. For example, the authors neglect to reference key publications: the papers from the Yoshikawa group about the discovery of mouse necdin, their necdin knockout mouse and work on neuronal abnormalities consequent to lack of necdin; the papers from the Wevrick group on the discovery of human necdin, the disruption of serotonergic axonal projections in necdin null mice, their discovery that serotonergic neurons in the hindbrain of necdin-null mice have abnormal cytoarchitecture and abnormal varicosities; the migration defect in necdin-deficient neurons; and the author's own work showing 5-HT abnormalities by HPLC (Zanella Advances in Experimental Medicine and Biology 2008) and the effect of fluoxetine in WT mice (Voituron, 2010). The authors should re-write these parts of the manuscript to better summarize key previous work, and put their new findings in this context.

[Editors' note: the manuscript was rejected after revisions, but the authors were invited to make a new submission.]

Thank you for submitting your work entitled "Necdin shapes serotonergic development and SERT activity modulating breathing in a mouse model for Prader-Willi Syndrome" for consideration by *eLife*. Your article has been reviewed by one of the original peer reviewers from the first submission, and the evaluation has been overseen by a Reviewing Editor and a Senior Editor. The reviewer opted to remain anonymous.

Our decision has been reached after consultation between the editors and the reviewer. Based on these discussions and the individual review below, we regret to inform you that your work will not be considered further for publication in *eLife* at this time. Because *eLife* does not permit two rounds of review, the editors have rejected this paper. If you and your co-authors feel you can address the issues as requested, you can then submit a new manuscript, which will be considered in light of the previous review.

The reviewers appreciate the importance of the work, and the insight provided by the pharmacological and genetic manipulations of the serotonin pathway to understand the modulation of breathing in this mouse model for Prader-Willi Syndrome, but there remained serious concerns that the text was not adequately revised to take into account the previous literature, and to correct errors in English. As described by the reviewer below, the use of a homozygous rather than in imprinted model is not sufficient to claim that the results are novel. With these concerns, and others raised by the reviewer, the team is concerned that readers will miss the importance of the work.

*Reviewer #2:*

The authors have responded to the original review primarily by adding a sentence at the end of each section describing how their work repeats and extends, to a greater or lesser extent, work previously done by others. Furthermore, they state that the work is all novel because they now use homozygous null mice rather than paternally inherited heterozygous mice. This makes the Results section very awkward and difficult to follow. It remains unclear which experiments provide truly novel data and which a confirmation of their work and that of others. Once the previously demonstrated findings are subtracted from the results, the novel results may become apparent. The manuscript can then be properly assessed for its suitability for publication, if the truly novel results are of sufficient interest. In summary, the authors need to rewrite the entire manuscript to describe what is already known about the serotonergic system in necdin +/- and necdin KO mice, then state their novel results.

The manuscript also has many awkward phrases that make it difficult to read. In the Abstract alone, they need to edit, for example:

· “Necdin-deficient mouse is the best relevant model” – awkwardly phrased.

· “Increase of 5-HT reuptake activity of the Serotonin Transporter (SERT)” – I don't think that was demonstrated by the experiments shown as they measured SERT transport in Figure 2, not reuptake.

· “Or fluoxetine treatment, a SERT inhibitor reuptake” – what is a SERT inhibitor reuptake?

· “Is sufficient to cause respiratory deficit in Necdin-KO pups” – this is vague.

· “SERT inhibition is a therapeutic perspective for PW patients” – what does therapeutic perspective mean, and why PW and not PWS is defined earlier in the same paragraph?

The rest of the manuscript has similar issues with awkwardly phrased and ungrammatical English that makes it difficult to interpret.

---

## [Author Response]

Essential revisions:1) The histological analysis does not indicate enough specifics on how quantification was performed, notably it does not appear that stereological methods were utilized. In the absence of such methods there are significant concerns for bias.

Effectively, we didn’t explain our method of quantification in the text. Now, we introduced this explanation in the Material and methods in the Immunohistochemistry and quantification part as following:

“Brainstem structures were sampled by selecting the raphe obscurus area and counting was performed on 3 sagittal sections of 100µm/animals which represent the entire PET1-YFP positive cell population of the raphe obscurus (ROb/B2) and pallidus (RPa/B1), both nuclei being difficult to separate. For each section, a Z-stack composed of 10 confocal images (8 µm focal spacing) was acquired. For quantification, stereological method has been applied on each Z-stack image using the eCELLence software developed by Glance Vision Technologies (Italy). The total cell number/ per animal was obtained by summing the sub-total of cells counted for the 3 Z-stacks.”

2) The authors focus on the serotonin system and the effect of modulating the 5-HT system genetically or pharmacologically. Previous studies have shown that other excitatory neuromodulators (Substance P and TRH) likewise partially rescue the respiratory deficit in brainstem-spinal cord preparations from Ndn-KO mouse pups (Pagliardini Advances in Experimental Medicine and Biology 2008). This raises the possibility that the serotonergic deficit is only one of many deficits in this genetic mutant. This point should be addressed and weighed in light of previous publications.

This point is extremely relevant. The reasons why, in regards to the respiratory distress, we studied the 5-HT neurons and system are: 1) previously we showed that Necdin is not expressed in the NKR1 positive neurons of the preBötzinger complex ((Zanella et al., 2008b); Figure S3), excluding a cell autonomous alteration of those cells; 2) Necdin is expressed in medullary 5HT neurons and Necdin deficiency in medullary preparations of *Ndn*-KO neonates alters the serotonergic modulation of the inspiratory rhythm generator (Zanella et al., 2008b), 3) Abnormal projections of 5-HT fibers toward both the rostral CNS and the RVLM were previously reported in *Ndn-*KO (Lee et al., 2005; Pagliardini et al., 2005). In addition, differences in the morphology of 5-HT vesicles from medullary fibers in sections performed at the level of the pre-Bötzinger complex were observed (Pagliardini et al., 2005; Zanella et al., 2008b). 4) Substance P, TRH and 5-HT are excitatory neuromodulators acting in part on the preBötzinger Complex to increase the frequency and stabilize inspiratory burst generation (Teran et al., 2015); both Substance P and TRH increase the frequency of inspiratory bursts and decrease the incidence of irregularly rhythm in brainstem-spinal cord preparations from *Ndn*-KO mouse pups (Pagliardini et al., 2005; Pagliardini et al., 2008). However, most substance P and TRH inputs to the preBötzinger complex come from the medullary 5-HT neurons (Hodges et al., 2008b; Holtman et al., 1994; Kachidian et al., 1991; Ptak et al., 2009).

While we agree with the reviewers that the 5-HT system is not the only one affected in *Ndn*-KO mice, the specific point raised here actually further emphasizes the critical role of 5-HT neuron alterations in driving respiratory rhythm perturbation in *Ndn*-KO mice. Furthermore our results also show that increase of extracellular 5-HT level alone is sufficient to restore eupneic breathing.

Consequently, in order to clarify this point, in the Introduction of our article, we included:

“Furthermore, it was previously shown that, in perinatal *Ndn*-KO mice in vitro, an irregular inspiratory rhythm and apneas resulted from an alteration in the regulation of the Inspiratory Rhythm Generator (IRG) (Ren et al., 2003; Zanella et al., 2008). 5-HT application, as well as other neuromodulators such as substance P and thyrotropin-releasing hormone (TRH), stabilized the inspiratory rhythm and reduced the incidence of apneas (Pagliardini et al., 2005; Zanella et al., 2008). Interestingly, 5-HT neurons co-release substance P and TRH, and represent the main source of these peptides to the IRG (Hodges et al., 2008; Holtman et al., 1994; Kachidian et al., 1991; Ptak et al., 2009). An anatomical study of the developing *Ndn*-KO medulla reported alterations in the majority of axonal tracts within the medulla and in particular an abnormal morphology and orientation of 5-HT and Substance P axonal fibers (Pagliardini et al., 2005; Pagliardini et al., 2008).”

3) The study would have much higher impact if the truly novel findings were disentangled from replication of previous results, although this replication is also important to report. The authors fail to place their experiments in the context of previously published work, in some cases contradicting their own previous studies (e.g. number of 5_HT neurons, (Zanella, 2008)) but confirming that of others (Pagliardini, 2005) with no reference to either study. This will put undue strain on readers to identify which are new observations and which are repeats of previously published observations.

Because we submitted the paper as a Short Report which has space limitation, we have not placed our experiments and results in the precise context of previously published work, including authors’ publications. Now, we introduced in our revised manuscript (see point 2) pioneer studies conducted in the Necdin mutant mouse that highlighted respiratory deficits and dysfunction of neuromodulation such as the 5-HT system (mainly by R. Wevrick’s and Muscatelli’s teams); however we knew neither the cause nor the consequences of this dysfunction. In our current study, our aim was to investigate the development and functional maturation of the 5-HT system since we first showed that Necdin is expressed in post-mitotic 5-HT precursors (in this paper) and throughout embryonic and postnatal development of 5-HT neurons. To conduct a rationale and complete investigation, we had to replicate experiments in the mouse model in order to make a complete story on deciphering causal link between 5-HT alteration and respiratory deficits in the same mouse model. Indeed some 5-HT linked alterations were previously described in different Ndn-KO models (different KO constructs, different genetic background, homozygous versus heterozygous mutants) and by different teams.

In this article we studied homozygous mutants (-/-) named here *Ndn*-KO. Necdin is an imprinted gene paternally expressed only, the maternal allele being silent. Previously we have shown that Necdin mice deleted for the paternal allele only (Ndn +m/-p) presented a milder phenotype than the *Ndn*-/- (deleted for both paternal and maternal alleles) due to a stochastic and faint expression of the maternal allele (Rieusset et al., 2013). This unexpected expression was the cause of a high variability in the severity of the phenotype that might result in not significant differences compared with the wild-type mice. To avoid this variability, we chose here to study the -/- *Ndn* mice.

To clarify this point we propose to add in the text, introduction of the Results/Discussion part:

“Necdin is an imprinted gene, paternally expressed only. In order to avoid a variability in our results due to a stochastic and faint expression of the maternal allele (Rieusset et al., 2013), here, we chose to work on *Ndn*-/- mice (named here *Ndn-*KO) instead of *Ndn* +m/-p mice as it has been done previously (Zanella et al., 2008).”

Importantly, the experiments we made in our first publication (Zanella et al., 2008) were performed on *Ndn*+m/-p mice named in the article *Ndn-*KO mice. Then in the review written by Zanella (Zanella et al., 2008b) based on the study published in Journal of Neurosciences, the *Ndn*-KO were cited as *Ndn* -/- instead of *Ndn*+m/-p; this was an error in the review.

If we consider the experiments performed on 5-HT neurons/system “which are repeats of previously published observations”, they are restricted to:

1) In the first paragraph entitled “Lack of Necdin affects the development and function of 5HT neurons”:

– Necdin expression in 5-HT neurons by immunohistochemistry (using a specific antibody validated on *Ndn*-/- brain sections) was already reported in Zanella et al. (Zanella et al., 2008). However here we precisely defined the extent of expression in the 5-HT neurons of the different raphe nuclei throughout embryonic and post-natal stages, showing for the first time that Necdin is expressed in 5HT post-mitotic precursors Figure 1—figure supplement 1.

– Alteration of 5HT projections have been observed by Pagliardini et al. (2008). This study is now cited in the Introduction. Here we observed a similar alteration illustrated in Figure 1—figure supplement 1. We modified the text as following:

“While 5-HT neuronal somas from rostral midbrain project theirs axons to the mesencephalon at E12.5, in *Ndn*-KO embryos, we observed a decrease in those ascending 5-HT projections (Figure 1—figure supplement 1) as previously described (Pagliardini et al., 2008).”

– The counting of 5-HT neurons in *Ndn*-KO pups (Figure 1).

Concerning the number of 5-HT neurons, to explain why “in some cases contradicting their own previous studies (e.g. number of 5-HT neurons, (Zanella, 2008) but confirming that of others (Pagliardini, 2005) with no reference to either study”. As explained above, *Ndn*+m/-p presented a milder phenotype than the *Ndn*-/- (deleted for both paternal and maternal alleles) due to a stochastic and faint expression of the maternal allele (Rieusset et al., 2013). In particular in this previous article (Rieusset et al., 2013), we showed that the number of 5HT neurons in the *Ndn*+m/-p is not significantly different compared to wild-type, as we already observed in the publication of Zanella et al., 2008. However, in *Ndn*-/- mice we observed a marked reduced number of 5-HT neurons (Rieusset et al., 2013). Here, by counting the Pet1eYFP+ neurons in *Ndn*-/- we confirmed the results and variability described in Rieusset et al., 2013. We clarified this point in the text as following:

“At E16.5, we observed a lack of organization of the 5-HT cytoarchitecture in *Ndn-*KO with misplaced 5-HT neurons leading in neonates to a significant ~30% reduction in the total number of 5HT-expressing neurons as quantified in the B1-B2 caudal Raphe nuclei (Figure 1); we already observed such a significant reduction in *Ndn*-/- mutants but not in *Ndn*+m/-p mutants (Rieusset et al., 2013).”

We modified the conclusion of this paragraph as following:

“Overall, our results showthat Necdin is critical for the establishment and functional features of 5-HT neurons during development.”

2) In the second paragraph entitled “Lack of Necdin increases the expression and activity of serotonin transporter SERT”:

– Neurite abnormalities with the presence of abnormal 5-HT vesicles/varicosities were observed in 5HT fibers (Pagliardini, 2005 and Zanella, 2008). In fact, we couldn’t prove that those abnormal 5-HT vesicles are either varicosities defined as “swellings that release their neurotransmitters at neuroeffector junctions” or spheroids defined as “dystrophic degenerative structures“. That is the reason why we named them here enLarged Punctiform Axonal (LPA) 5-HT staining. Furthermore, for the first time, we quantified them in different brain regions and in different genetic mutant mice (Ndn KO, Sert KO, Ndn/Sert DKO). We modified the text as following:

“An alteration of 5-HT metabolism was already observed in *Ndn*-KO mice (Zanella et al., 2008), along with abnormal 5-HT fibers with “large varicosities “(Pagliardini et al., 2005; Zanella et al., 2008). Using immunohistochemistry (IHC), we quantified these enlarged punctiform axonal (LPA) 5-HT staining, showing a significant increase in different *Ndn-*KO brain regions compared with WT (Figure 2).”

3) In the third paragraph entitled “Genetic ablation or pharmacological inhibition of SERT uptake restores normal breathing in *Ndn-*KO mice”.

– We reported in a previous publication that in wild type mice: “Fluoxetine Treatment Abolishes the in vitro Respiratory Response to Acidosis in Neonatal Mice” (Voituron et al., 2010). Here, since we investigated the effect of fluoxetine in *Ndn*-KO, the experiments in control animals was mandatory.

We modified the text as following:

“Looking at the chemosensitive response to central acidosis in *en bloc* brainstem-spinal cord, we found that fluoxetine treatment abolished the capacity of WT to respond to acidosis as we previously observed (Voituron et al., 2010) (Figure 4—figure supplement 2).”

All the other experiments of this paper “are new published observations”.

4) The abnormality of serotonergic function in the respiratory system in the hindbrain of Ndn-null mice has been studied by several groups over the last decade, compromising the novelty of the study. For example, the authors neglect to reference key publications: the papers from the Yoshikawa group about the discovery of mouse necdin, their necdin knockout mouse and work on neuronal abnormalities consequent to lack of necdin; the papers from the Wevrick group on the discovery of human necdin, the disruption of serotonergic axonal projections in necdin null mice, their discovery that serotonergic neurons in the hindbrain of necdin-null mice have abnormal cytoarchitecture and abnormal varicosities; the migration defect in necdin-deficient neurons; and the author's own work showing 5-HT abnormalities by HPLC (Zanella Advances in Experimental Medicine and Biology 2008) and the effect of fluoxetine in WT mice (Voituron, 2010). The authors should re-write these parts of the manuscript to better summarize key previous work, and put their new findings in this context.

We agree with this comment. This point is somehow linked to point 2 and 3 and the modifications made to answer those both points should also be considered in point 4. Indeed, we tried to introduce all the key previous work in different parts of the text: Introduction, Results and Discussion. All the studies and publications cited by the reviewers have been now introduced. The publication (book review) of Zanella et al. in Advances in Experimental Medicine and Biology 2008 has been replaced by the original study: Zanella et al., 2008.

[Editors' note: what now follows is the author responses after they submitted for further consideration.]

[…] Reviewer #2:The authors have responded to the original review primarily by adding a sentence at the end of each section describing how their work repeats and extends, to a greater or lesser extent, work previously done by others.

We actually modified considerably all parts of the manuscript. We introduced and discussed previous literature, in order to better contextualize our results. In the Introduction, we extended the description of breathing deficits in PWS patients, and the previous results obtained on *Ndn*-KO mice linked to the breathing deficits. We also extended the conclusion, starting by summarizing the previous results published on Necdin, 5-HT and breathing in order to discriminate our novel results from those previous results. We explained how and when Fluoxetine might be used in PWS patients to alleviate their breathing deficits. We also modified the Material and methods section providing more explanations on the use of Malgel2-/- mice (see next paragraph). In the present version, we have also included 25 new references (compared to the initial version). In the previous version, there were some novel results described in the figure legends but not in the main text. In the present version, we thus transferred this information from the figure legends to the result part.

Furthermore, they state that the work is all novel because they now use homozygous null mice rather than paternally inherited heterozygous mice.

We never made such a claim and we apologize for this misunderstanding: the novelty aspect of our study is discussed compared to the literature in the Results/Discussion and conclusion sections, and never refers to the use of heterozygous vs. homozygous mice. The use of homozygous mice allowed us to reduce the variability in the severity of their phenotype (apnea). Indeed, we previously observed that heterozygous mice show an important phenotypic variability, ranging from strong to light phenotypic alteration, due to a stochastic expression of the maternal allele (Rieusset et al., 2013). Here, the use of homozygous mice is justified scientifically and ethically, as it allowed us to get significant results using a smaller number of mice, since their phenotype (apnea in particular) was of greater homogeneity. We explained why we used homozygous rather than heterozygous mice in the “Material and Methods” part, which is the most appropriated section, avoiding any misunderstanding concerning the novelty of our results (such as considering that the novelty is the use of Ndn-/- mice).

It remains unclear which experiments provide truly novel data and which a confirmation of their work and that of others. Once the previously demonstrated findings are subtracted from the results, the novel results may become apparent.

In order to clarify this point, we made a table listing all figures (see table below) indicating which precisely are the repeated and novel experiments, the references of the repeated experiments and the justification to repeat them. Less than 10% of experiments have been replicated, consequently more than 90% are novel.

The manuscript can then be properly assessed for its suitability for publication, if the truly novel results are of sufficient interest. In summary, the authors need to rewrite the entire manuscript to describe what is already known about the serotonergic system in necdin +/- and necdin KO mice, then state their novel results.

Indeed, 90% of the results presented in this article are novel results that have never been published in neither heterozygous nor homozygous *Necdin*-KO mice. In the new version, novel results are clearly stated in the Abstract and in the conclusion (“Here, we showed…healthy mice.”). In the “Results and Discussion” part we also present our results with respect to the previous ones.

Overall, we hope that the readers will see the novelty of our results, listed below:

– The expression of Necdin in early post-mitotic precursors (expressing Pet1) followed by an expression in all 5-HT neurons;

– The misplacement and disorganization of 5-HT raphe nuclei in *Ndn*-KO mice. The decrease in the number of 5-HT neurons in the B1-B2 nuclei, a result not previously observed in our *Ndn*+m/-p mice but observed in *Ndn*-/- mice and we explained why (see table);

– An in vivo alteration in cell migration of 5-HT precursors;

– An alteration of firing properties of 5-HT neurons recorded on brain slices of mutant;

– The quantification of enlarged Punctiform Axonal (LPA) 5-HT stainings in different brain regions and in different genetic backgrounds (WT, *Ndn*-KO, *Slc6a4* and *Ndn/Slc6a4*-DKO) showing the lack of Sert expression is sufficient to suppress the 5-HT LPA stainings;

– The overexpression of Sert resulting from a post-transcriptional or post-traductional regulation and resulting in an increase activity of SERT transport and uptake (indeed the experiment using ASP+ allows the visualization and quantification of ASP+ uptake through SERT (Lau et al., 2015));

– The genetic ablation of *Slc6a4* (encoding SERT) in *Ndn*-KO (*Ndn/Slc6a4*-DKO) leading to a rescue of the phenotype (normalization of apnea and loss of accumulation of 5-HT in LPAs);

– The in vivo positive effect of early Fluoxetine treatment in *Ndn*-KO mice allowing apnea to be normalized;

– The in vivo adverse effect of early Fluoxetine treatment in WT mice, inducing apnea. We are not aware of any published or unpublished similar results and so considered that they are novel.

The manuscript also has many awkward phrases that make it difficult to read.

We are really sorry for the grammatical errors (this article was written by French scientists). The new manuscript has been read and corrected by Pr. Keith Dudley (a native English scientist) and Dr Simon McMullan (Australian Senior scientist).

In the Abstract alone, they need to edit, for example:“Necdin-deficient mouse is the best relevant model” – awkwardly phrased.

Now replaced by: *“*The *Necdin*-deficient ouse is the only model…”.

“Increase of 5-HT reuptake activity of the Serotonin Transporter (SERT)” – I don't think that was demonstrated by the experiments shown as they measured SERT transport in Figure 2, not reuptake.

We agree that the reuptake is not appropriated. However uptake is the appropriate word since we measure in Figure 2 (and Figure 2—figure supplement 2) the SERT uptake activity using ASP+ as fluorescent substrate and we measured the kinetic parameters of Asp+ uptake in primary cultures of raphe neurons. Indeed those experiments measure SERT transport activity and the uptake activity (Lau et al., 2015).

“Or fluoxetine treatment, a SERT inhibitor reuptake” – what is a SERT inhibitor reuptake?

This was a big mistake that we didn’t see. Of course it has been corrected by: a 5-HT reuptake inhibitor.

“Is sufficient to cause respiratory deficit in Necdin-KO pups” – this is vague.

This has been replaced by: is sufficient to cause the apneas in *Necdin-*KO pups.

“SERT inhibition is a therapeutic perspective for PW patients” – what does therapeutic perspective mean, and why PW and not PWS is defined earlier in the same paragraph?

This has been replaced by:and that Fluoxetine opens a novel perspective to improve ventilator control in patients with PWS.

The rest of the manuscript has similar issues with awkwardly phrased and ungrammatical English that makes it difficult to interpret.

We have modified the Abstract, and the text has been proofread and corrected by Pr. Keith Dudley (a native English speaking scientist) and Dr. Simon McMullan (Australian Senior scientist), as mentioned above.

Experiments: referenced figuresNovel or Replicate experiment (article reference if replication)CommentsFigure 1NovelExpression of Necdin in 5-HT neuronal precursors has never been shownFigure 1NovelSuch disorganization when the 5-HT nuclei should be set up was not reported in any previous publication.Figure 1Replicate (Zanella et al., 2008; Rieusset et al; 2013)The number of 5-HT neurons has been counted in the *Ndn -/-* mice studied in this article. Previously (Zanella, 2008) we showed that in *Ndn+m/-p* mice, the number of 5-HT neurons was not significantly reduced compared with WT. However we showed that Ndn +/-p mice present a high variability in the severity of the phenotype, cellular defects due to a stochastic and faint expression of the maternal allele in some mice. To avoid this variability we chose to study *Ndn-*/- mice.Figure 1NovelIt’s the first time that defects in radial migration of *Ndn*-deficient 5-HT precursors are shown *in vivo* using twophoton time-lapse imaging. Furthermore, our study revealed a problem of somal translocation. Problems of migration of other neuronal populations in *Ndn-*KO mice have been suggested but never shown *in vivo*.Figure 1NovelFigure 1NovelFigure 1NovelFigure 1NovelFigure 1NovelElectrophysiological recordings of 5-HT neurons in *Ndn-KO* mice have never been reported.Figure 1NovelFigure 1NovelFigure 1—figure supplement 1NovelThis panel shows (for the first time) the expression of Necdin in Pet-EYFP 5-HT precursors at E10.5. Then we show that Necdin is expressed in all 5-HT neurons in raphe nuclei. Again, the expression of Ndn by immunohistochemistry in all 5-HT raphe nuclei has never been shown (only suggested in Zanella).Figure 1Replicate (Lee et al., 2005)Indeed, we looked at the distribution of 5-HT neurons in *Ndn*-KO mice at nearly all development stages; we chose to show the E12.5 and E16.5 because they suggest a delay in migration. Previously, Lee et al. showed a similar defect at E12.5, in their *Ndn*- KO model, proposing a defect in serotonergic projections reflecting a more general delay or dysfunction of axonal projections. We think that it is important to show that this defect is present in both *Ndn*-KO mouse models.Figure 2NovelThe abnormal 5-HT fibers with “large varicosities “were previously described (Pagliardini et al., 2005; Zanella et al., 2008). The novelty here is: the quantification in different brain structures of the 5-HT LPAs and, importantly the fact that we show that those 5-HT LPAs disappear in *Ndn/Slc6a4*-DKO (normal labelling being present in *Slc6a4*-KO mice).Figure 2All NovelsThe quantification of SERT in *Ndn*-KO has never been done until this study.Figure 2All NovelsASP+uptake assays in *Ndn*-KO serotonergic cells (primary cultures of raphe) compared with WT and SERT-KO (negative control) cells have never been done previously. This experiment allows to assess the activity of SERT as a transporter leading to an uptake of ASP+.Figure 2—figure supplement 1NovelL-Trp quantification (5-HT substrate) has never been done in Ndn-KO mice.Figure 2—figure supplement 1Replicate (Zanella et al., 2008)5-HT and 5-HIAA were previously quantified in *Ndn*+m/-p. It was necessary to quantify them here in the *Ndn*-/- animals and to include those quantifiactions in a more complete analysis of the 5-HT system.Figure 2—figure supplement 1Novel*Tph2* transcripts have not been quantified previously.Figure 2—figure supplement 1All Novels*Slc6a4* (Sert) tanscripts have not been quantified previously.Figure 2—figure supplement 2All NovelsIllustration of the ASP+ uptake in the 5- HT neurons.Figure 3All novelsPlethysmography recordings and breathing parameters analysis of *in vivo* Fluoxetine treatment in early life of Necdin deficient mice or Ndn/Sl6a4 DKO mice in comparison with Ndn-KO or WT mice. Recordings are performed at P15 (O DAT) after a 10 days treatment with Fluoxetine.Figure 3—figure supplement 1All novelsSimilar experiments as performed in Figure 3 but performed at 15 DAT and 30 DAT in order to assess a mid/long term effect of fluoxetine treatment.Figure 3—figure supplement 2All novelsFigure 4All novels*In vivo*
**hypercapnia** and *in vitro* acidosis have never been assessed either in *Ndn*KO mice or in *Ndn-*KO brainstem-spinal cord preparations treated or not with fluoxetine. In our previous study (Zanella et al., 2008) **hypoxia** only was assessed.Figure 4—figure supplement 1All novels8-OH-DPAT on *Ndn*-KO brainstem-spinal cord *in vitro* preparations has never been assessed previously.Figure 4—figure supplement 2Novel experiment: treatment with 8-OHDPAT Replicate experiment: acidosis in WT preparations treated or not with fluoxetine (Voituron et al., 2010)In 2010, Voituron et al. showed that “Fluoxetine Treatment Abolishes the In Vitro Respiratory Response to Acidosis in Neonatal Mice”. In our study, since we investigated the effect of fluoxetine in *Ndn*-KO brainstem-spinal cord preparations submitted to acidosis, the experiments in control animals were mandatory and are presented in this figure. Here, we added a novel experiment in control mice: the effect of 8-OH-DAPT.